# CAUSAL-COT: CAUSAL CHAIN-OF-THOUGHT FOR VALIDATED REASONING

## ABSTRACT

Chain-of-Thought (CoT) prompting enables large language models (LLMs) to expose intermediate reasoning, but the resulting rationales are often unfaithful—skipping premises, confusing relations, or relying on unsupported leaps. We propose **Causal-CoT**, a framework that integrates causal graph construction, augmentation, and verification into the CoT paradigm. Causal-CoT operates through a three-stage pipeline: (1) *DAG-guided CoT* constructs an initial causal graph from the problem context; (2) *Reflection and Augmentation* enriches the graph by adding plausible mediators and contextual variables; and (3) *Causal Verification* estimates conditional probabilities via prompting and applies do-calculus to compute causal effects. This structured approach transforms linear reasoning into graph-based inference, enabling more faithful and interpretable reasoning. Experiments across seven benchmarks in mathematics, commonsense, and causal reasoning show that Causal-CoT improves reasoning fidelity, mitigates shortcut behaviors, and achieves more stable performance compared to standard CoT. Moreover, Causal-CoT significantly enhances both reasoning fidelity and answer accuracy, effectively suppresses "jump-to-answer" shortcuts, and strikes a favorable balance between accuracy and computational cost.

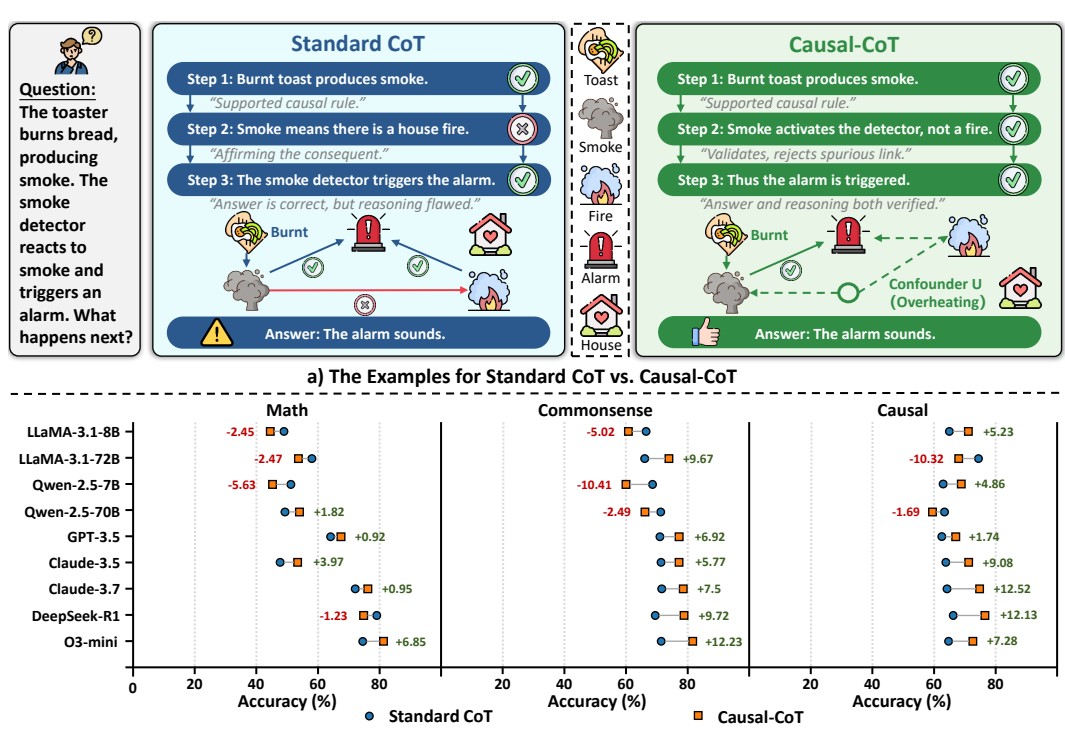

Figure 1: The results for Standard CoT vs. Causal-CoT.

# 1 INTRODUCTION

Large Language Models (LLMs) have demonstrated remarkable performance in diverse reasoning tasks, including mathematics (Cobbe et al., 2021; Hendrycks et al., 2021), commonsense reasoning (Putri et al., 2024; Rein et al., 2023), and causal reasoning (Chi et al., 2025; Wang et al., 2024). This success is largely driven by Chain-of-Thought (CoT) prompting ((Wei et al., 2022; Kojima et al., 2022; Wang et al., 2023a)), which encourages models to generate step-by-step rationales before arriving at a final answer, thereby externalizing intermediate reasoning to tackle complex problems.

Despite these advantages, recent studies (Paul et al., 2024; Li et al., 2024; Tanneru et al., 2024; Chi et al., 2025) reveal a key limitation: ***while final answers may be correct, the intermediate steps are often unverifiable, logically inconsistent, or causally unsound***. This allows LLMs to arrive at accurate outcomes through spurious reasoning, eroding interpretability and reliability—particularly in domains like scientific discovery, decision-making, and safety-critical applications.

One promising direction for mitigating these issues is to impose causal structure on the reasoning process. Causality provides tools for representing directional dependencies, distinguishing genuine causal effects from correlations, and testing claims through interventions (Pearl, 2009; Imbens & Rubin, 2015). However, effective causal verification requires a complete causal graph and precise effect estimation. First, incomplete graphs (e.g., lacking critical variables or relationships) yield unreliable inferences (Zanga et al., 2022; Kiciman et al., 2023), and existing causal discovery methods (Vowels et al., 2022; Kaltenpoth & Vreeken, 2023; Ma, 2024) struggle with unmeasured confounders issue. Second, causal effect estimation via do-calculus needs distributions of intervention, which are limited to data structure. We address these challenges by leveraging LLMs to (1) improve graph completeness through CoT prompts that elicit missing premises, mediators, or confounders, and (2) estimate causal effects using token-level conditional probabilities (e.g., $p$("A occurs" | "B holds")), since LLMs compute a probability distribution over the vocabulary for each token conditioned on the semantic meaning of preceding context and their internalized knowledge, thereby serving as a bridge between causal inference and world knowledge.

We propose **Causal-CoT**, a novel framework that integrates causal graph construction, augmentation, and verification into the CoT paradigm. Unlike standard CoT, which produces linear and opaque rationales, Causal-CoT maps reasoning to a Directed Acyclic Graph (DAG), refines this structure, and applies do-calculus causal verification (see Figure1-a). The framework operates through three core components: (1) *DAG-guided CoT*, which constructs an initial causal structure of nodes and edges, and (2) *CoT Reflection and Augmentation*, which enhances the graph via prompt-based refinement or information-retrieval-based enrichment. (3) *Causal Verification* leverages do-calculus with LLM-derived probabilities. As demonstrated in Figure1-b), Causal-CoT significantly outperforms standard CoT across seven benchmarks in mathematics, commonsense, and causal reasoning, achieving enhanced answer accuracy and reasoning fidelity. Our results show that adding causal structure to CoT reasoning improves reasoning quality and stability across diverse LLMs. Our work makes three main contributions:

- We introduce **Causal-CoT**, a three-stage framework that transforms linear CoT rationales into graph-structured causal reasoning through DAG-guided CoT, graph augmentation, and do-calculus–based causal verification.

- We formalize reasoning fidelity using two complementary gaps: accuracy $\Delta_r$ and reasoning fidelity $\Delta_s$, enabling a clear separation between structural and knowledge-level errors.

- Through extensive experiments on nine LLMs across eight benchmarks, we show that Causal-CoT improves reasoning fidelity, mitigates shortcut behaviors, and offers more stable accuracy-efficiency trade-offs than CoT, while analyzing when causal structuring helps or hurts.

# 2 CAUSAL-COT

The **Causal-CoT** framework enhances CoT reasoning by integrating fundamental principles from causal inference, transforming linear rationales into a structured, verifiable causal reasoning pipeline. As depicted in Figure 2, Causal-CoT guides graph construction, augmentation, and verification, ultimately yielding more robust and interpretable reasoning.

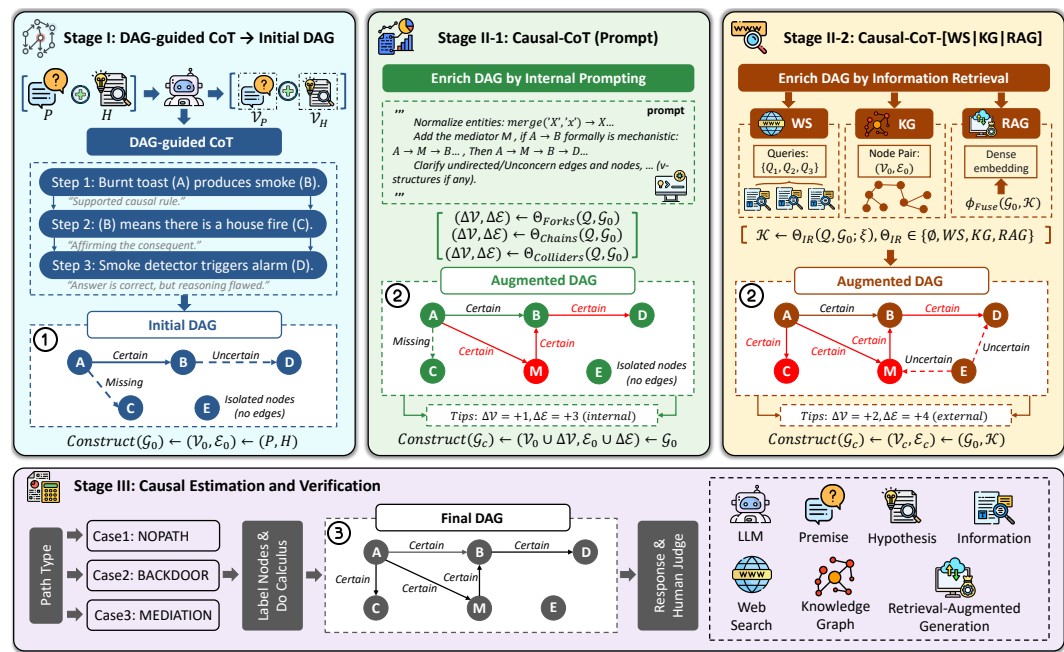

Figure 2: Unified pipeline of **Causal-CoT**. Stage I constructs an initial DAG. Stage II enriches the DAG through prompt-based augmentation or information retrieval (IR) from web, KG, or RAG. Stage III performs causal verification via causal do-calculus.

## 2.1 CAUSAL PRELIMINARIES

Causal-CoT builds on the formalism of Structural Causal Models (SCMs). An SCM defines a set of endogenous variables $\mathcal{V}$ through structural assignments:

$$V_i = f_i(\mathrm{Pa}(V_i), U_i), \tag{1}$$

where $\mathrm{Pa}(V_i)$ are the direct causes of $V_i$ and $U_i$ are exogenous disturbances. These equations induce a directed acyclic graph (DAG):

$$\mathcal{G} = (\mathcal{V}, \mathcal{E}), \qquad (X \to Y) \in \mathcal{E} \iff X \in \mathrm{Pa}(Y), \tag{2}$$

which encodes asymmetric causal dependencies. Different node roles follow directly from the graph: confounders ($Z \to X$ and $Z \to Y$), mediators ($X \to M \to Y$), and colliders ($X \to Z \leftarrow Y$). Central to SCMs is the intervention operator:

$$P(Y \mid do(X = x)) = P(Y \mid f_X(\cdot) \text{ is replaced by the constant assignment } X := x). \tag{3}$$

which removes incoming edges to $X$ and evaluates how forcing $X$ changes the distribution of $Y$. The existence of a causal effect is determined by

$$\mathrm{ATE}(X \to Y) = E(Y \mid do(X{=}1)) - E(Y \mid do(X{=}0)), \tag{4}$$

with its sign and magnitude indicating causal strength. These notions motivate the structure of Causal-CoT. A natural-language input must be mapped to variables, directed relations, and candidate pathways (via a DAG); enriched with additional structural hypotheses; and finally validated through interventional-style probability queries.

## 2.2 STAGE I: DAG-GUIDED COT

The CoT output is formalized into a premise and a hypothesis, and then the premise is broken into small atomic statements. Each statement is a node. Edges link nodes according to the relations given in the problem. Formally, given a premise $P$ and hypothesis $H$, we first decompose them into atomic statements $\phi_{\mathrm{ATOM}}$: $\mathcal{V}_P = \{\phi_{\mathrm{ATOM}}(s) : s \in \mathrm{DECOMPOSE}(P)\}$, $\mathcal{V}_H = \{\phi_{\mathrm{ATOM}}(s) : s \in \mathrm{DECOMPOSE}(H)\}$. Let $\mathcal{V}_0 = \mathcal{V}_P \cup \mathcal{V}_H$. The initial DAG is

$$\mathcal{G}_0 = (\mathcal{V}_0, \mathcal{E}_0) = \phi_{\mathrm{CONSTRUCT}}(P, H), \tag{5}$$

where an edge $(u \rightarrow v) \in \mathcal{E}_0$ encodes the claim that "$u$ is a premise/cause of/ related to $v$". The initial DAG shows the problem's structure and points out the causal question $Q = (X \rightarrow Y)$, but adds no extra edges beyond what the problem states.

## 2.3 STAGE II: COT AUGMENTATION AND REFLECTION.

In the augmentation stage, the DAG is refined with knowledge-based reasoning to form a complete causal graph. All paths from $X$ to $Y$ are recognized. To be specific, starting from the initial graph $\mathcal{G}_0 = (\mathcal{V}_0, \mathcal{E}_0)$, we refine it into $\mathcal{G}_c = (\mathcal{V}_c, \mathcal{E}_c)$ using domain knowledge and correlations, ensuring acyclicity. From the query $Q$ and base graph $\mathcal{G}_0$, we add forks for latent common causes, chains for mediators, and colliders for v-structures, then combine all justified updates to form the completed graph $\mathcal{G}_c = (\mathcal{V}_0 \cup \Delta\mathcal{V}, \ \mathcal{E}_0 \cup \Delta\mathcal{E})$, where

$$(\Delta\mathcal{E}, \Delta\mathcal{V}) = \begin{cases} \Phi_{\text{Forks}}(Q, \mathcal{G}_0), & \Delta\mathcal{E} = \{(U \rightarrow X), (U \rightarrow Y)\}, & \text{if } U \text{ is a latent common cause,} \\ \Phi_{\text{Chains}}(Q, \mathcal{G}_0), & \Delta\mathcal{E} = \{(X \rightarrow M), (M \rightarrow Y)\}, & \text{if } M \text{ mediates } X \rightarrow Y, \\ \Phi_{\text{Colliders}}(Q, \mathcal{G}_0), & \Delta\mathcal{E} = \{(X \rightarrow Z), (Z \leftarrow Y)\}, & \text{if } Z \text{ is a collider of } X, Y. \end{cases} \quad (6)$$

To refine the initial DAG $\mathcal{G}_0$, we employ two complementary augmentation strategies. (1) *Prompt-based augmentation* leverages the LLM's internal knowledge to expand nodes and edges. (2) *IR-based augmentation* addresses uncertain or incomplete edges by converting them into structured queries and retrieving external evidence from Web search, knowledge graphs, or domain-specific RAG stores. Formally, the retrieval operator is defined as

$$\mathcal{K} = \Phi_{\text{IR}}(Q, \mathcal{G}_0; \xi), \qquad \Phi_{\text{IR}} \in \{\varnothing, \text{WEB}, \text{KG}, \text{RAG}\}, \quad (7)$$

where $\mathcal{K}$ denotes the evidence pool and $\xi$ specifies the retrieval budget (e.g., top-$k$ snippets or paths). A fusion operator integrates retrieved evidence with the LLM-augmented graph:

$$\mathcal{G}_c = (\mathcal{V}_c, \mathcal{E}_c) = \phi_{\text{FUSE}}(\mathcal{G}_0, \mathcal{K}). \quad (8)$$

## 2.4 STAGE III: CAUSAL ESTIMATION AND VERIFICATION.

After completing the graph, the model (via prompts) classifies the $X-Y$ linkage as (i) no confounder/mediator path, (ii) a backdoor path, or (iii) a mediation path, labeling intervening nodes as confounders, mediators, or colliders. For causal effect estimation, each required conditional probability is obtained with a single targeted prompt that queries the semantic relationship to derive its likelihood, which is then plugged into the causal effect computation formula. Finally, a fixed threshold determines whether to accept $X \rightarrow Y$ as causal.

Under the ignorability assumption(Rubin, 1974; Pearl, 2010), the causal effect

$$\text{ATE} = \mathbb{E}[Y|do(X=1)] - \mathbb{E}[Y|do(X=0)] \quad (9)$$

can be identified and estimated by plugging in observed conditional probabilities. Here $Y$ is binary (e.g., "is A" / "not A" in semantics), and the effect of $X \rightarrow Y$ is computed from the three paths.

- **Case 1: No Confounder and Mediator.**

$$\text{ATE} = p(Y|X=1) - p(Y|X=0). \quad (10)$$

- **Case 2: Backdoor Adjustment.** With a confounder set $\mathcal{Z}$, if $\mathcal{Z} \neq \varnothing$:

$$\text{ATE} = \sum_{z \in \mathcal{Z}} \Big( p(Y|X=1, z) - p(Y|X=0, z) \Big) p(z). \quad (11)$$

- **Case 3: Mediation.** With multiple mediator paths $\{\mathcal{M}_1^*, \mathcal{M}_2^*, \dots\}$, only the first mediator of each path is considered, where $\Lambda(*) = p(Y \mid X = *, M_k = m, z)$ for $* = 0 \mid 1$:

$$\text{NDE} = \sum_{z \in \mathcal{Z}} p(z) \sum_k \sum_{m \in \{0,1\}} \Big[ \Lambda(1) - \Lambda(0) \Big], p(M_k = m \mid X = 0, z) \quad (12)$$

---

**Algorithm 1 Causal-CoT (DAG-guided CoT → Reflection/IR → Causal Verification)**

---

**Require:** Premise $P$, Hypothesis $H$; LLM $\mathcal{M}$; node set $\mathcal{V}$; initial edges $\mathcal{E}_0$; augmentation mode $\Phi_{\text{AUG}} \in \{\text{PROMPT}, \text{IR}\}$; retrieval operator $\Phi_{\text{IR}} \in \{\text{KG}, \text{WEB}, \text{RAG}\}$; budget $\xi$

**Ensure:** *Judge_result* $\in \{\text{TRUE}, \text{FALSE}\}$, validated CPDAG $\mathcal{G}^{\star}$

    **Stage I. DAG-guided CoT**

1:  $\mathcal{V}_0 \leftarrow \{\phi_{\text{ATOM}}(s) : s \in \text{DECOMPOSE}(P) \cup \text{DECOMPOSE}(H)\}$

2:  $\mathcal{E}_0 \leftarrow \text{EXTRACTSTATEDRELATIONS}(P)$

3:  $\mathcal{G}_0 \leftarrow \phi_{\text{CONSTRUCT}}(\mathcal{V}_0, \mathcal{E}_0)$

4:  $Q = (X, Y) \leftarrow \text{EXTRACTCAUSALQUESTION}(P, H); \quad Q \notin \mathcal{E}(\mathcal{G}_0)$

    **Stage II. CoT Augmentation & Reflection**

5:  **if** $\Phi_{\text{AUG}} = \text{IR}$ **then**

6:     $\mathcal{K} \leftarrow \emptyset$

7:     **for** $e \in \text{UNCERTAINEDGES}(\mathcal{G}_0; Q)$ **do**

8:        $Q_e \leftarrow \text{FORMULATEQUERY}(Q, e); \quad \mathcal{K} \leftarrow \mathcal{K} \cup \text{RETRIEVE}(Q_e; \Phi_{\text{IR}}, \xi)$

9:     **end for**

10:    $\mathcal{S} \leftarrow \text{FILTERDEDUPRANK}(\mathcal{K})$

11:  **else**

12:    $\mathcal{S} \leftarrow \text{INTERNAL}$

13:  **end if**

14:  $(\Delta\mathcal{V}, \Delta\mathcal{E}) \leftarrow \sum_{\psi \in \{\phi_{\text{FORKS}}, \phi_{\text{CHAINS}}, \phi_{\text{COLLIDERS}}\}} \psi(Q, \mathcal{G}_0; \mathcal{S})$

15:  $\mathcal{G}_c \leftarrow \phi_{\text{FUSE}}(\mathcal{G}_0, \Delta\mathcal{V}, \Delta\mathcal{E}; \text{prov} = \mathcal{S})$

16:  $\text{ENFORCEACYCLIC}(\mathcal{G}_c); \quad Q \notin \mathcal{E}(\mathcal{G}_c)$

    **Stage III. Causal Estimation & Verification**

17:  $\Pi \leftarrow \text{PATHTYPE}(\mathcal{G}_c, Q)$                 ▷ $\Pi \in \{\text{NOPATH}, \text{BACKDOOR}, \text{MEDIATION}\}$

18:  $(\mathcal{Z}, \mathcal{M}^*) \leftarrow \text{LABELNODES}(\mathcal{G}^{\star}, Q, \Pi)$             ▷ confounders & mediators

19:  **Assume** ignorability; treat $Y$ as binary; use LLM token probs $p(\cdot)$

20:  $\widehat{\delta} \leftarrow \begin{cases} \text{case 1: } \widehat{\text{ATE}} \text{ from Eq.10,} & \Pi = \text{NOPATH} \\ \text{case 2: } \widehat{\text{ATE}} \text{ from Eq.11,} & \Pi = \text{BACKDOOR} \\ \text{case 3: } \widehat{\text{TE}} \text{ from Eq.12,} & \Pi = \text{MEDIATION} \end{cases}$

21:  *Judge_result* $\leftarrow [\widehat{\delta} > \tau]$

22:  **return** *Judge_result*, $\widehat{\delta}$, $\Pi$, $\mathcal{G}^{\star}$

---

$$\text{NIE} = \sum_{z \in \mathcal{Z}} p(z) \sum_k \sum_{m \in \{0,1\}} \Lambda(0) \Big[ p(M_k = m \mid X = 1, z) - p(M_k = m \mid X = 0, z) \Big], \quad (13)$$

$$\text{TE} = \text{NDE} + \text{NIE}, \quad (14)$$

If ATE or TE $> \tau$, then $X$ is judged to have a significant causal effect on $Y$. Deriving conditional-probability likelihoods from an LLM provides a practical workaround for two obstacles in causal inference: (1) LLMs have limited causal reasoning capability (Jin et al., 2023a; Chi et al., 2024; Jin et al., 2023b), since they are trained on correlations and direct effect estimation is therefore unreliable, and (2) classical methods face a data bottleneck because interventional conditionals are often unavailable. Instead of directly taking numeric probabilities, we use a single targeted prompt (See appendix.E.8) to query the LLM with semantic conditionals (e.g., "How likely is event $A$ given context $B$?"), map its verbal likelihoods (*very unlikely*, *unlikely*, *possible*, *likely*, *very likely*) to a calibrated probability distribution, and use the distributional mean as the conditional probability substituted into the causal formulas.

**Algorithmic Formalization.** Algorithm 1 presents the unified procedure. (i) use prompting to construct an initial DAG, (ii) enrich it through prompt-based or IR-based augmentation, and (iii) perform causal verification via do-calculus using a single targeted prompt to obtain likelihood estimates. This formalization establishes Causal-CoT as a modular and extensible framework unifying reasoning across mathematics, commonsense, and causal inference tasks.

Table 1: **Main results for Causal-CoT (Accuracy %).** The upper block reports baseline averages across nine LLMs. The lower block reports per-model Causal-CoT results, with green / red indicating deltas over CoT. Best per-column value is **bold-underline**.

| Method / Model | Causal | | Math | | Commonsense | | Avg. |
|---|---|---|---|---|---|---|---|
| | CAUSALNET | E-CARE | MATH | AIME | CSQA | GPQA | |
| **Baselines** | | | | | | | |
| Zero-shot | 48.30 | 58.44 | 34.12 | 38.20 | 47.90 | 27.65 | 42.44 |
| Few-shot | 52.48 | 62.15 | 41.26 | 42.11 | 53.17 | 26.93 | 46.35 |
| CoT ($\Delta_\dagger$) | 61.20 | 80.77 | 49.33 | 61.62 | 69.89 | 38.02 | 60.14 |
| ToT | 62.47 | 78.35 | 53.62 | 62.84 | 71.28 | 44.53 | 62.18 |
| **Open-source LLMs (Causal-CoT)** | | | | | | | |
| LLaMA-3.1-8B$^{(\Delta_r)}$ | 71.43$^{(+5.2)}$ | 81.36$^{(-1.1)}$ | 54.71$^{(+2.4)}$ | 45.45$^{(-2.4)}$ | 64.90$^{(-5.0)}$ | 55.36$^{(+14.8)}$ | 62.87$^{(+2.7)}$ |
| LLaMA-3.1-70B$^{(\Delta_r)}$ | 50.90$^{(-10.3)}$ | 85.93$^{(+3.0)}$ | 42.10$^{(-7.2)}$ | 55.45$^{(-2.4)}$ | 79.50$^{(+9.6)}$ | 58.31$^{(+20.3)}$ | 62.36$^{(+2.2)}$ |
| Qwen2.5-7B$^{(\Delta_r)}$ | 67.20$^{(+4.8)}$ | 66.04$^{(-8.3)}$ | 59.90$^{(+10.6)}$ | 45.10$^{(-5.6)}$ | 59.50$^{(-10.4)}$ | 54.40$^{(+16.4)}$ | 58.69$^{(-1.5)}$ |
| Qwen2.5-72B$^{(\Delta_r)}$ | 59.60$^{(-1.6)}$ | 80.70$^{(+3.0)}$ | **61.90**$^{(+5.6)}$ | 47.37$^{(+1.8)}$ | 68.30$^{(-2.4)}$ | 63.92$^{(+25.9)}$ | 63.30$^{(+3.2)}$ |
| **Closed-source LLMs (Causal-CoT)** | | | | | | | |
| GPT-3.5$^{(\Delta_r)}$ | 57.90$^{(-3.3)}$ | 77.19$^{(+1.7)}$ | 53.30$^{(+4.0)}$ | 69.64$^{(+0.9)}$ | 76.80$^{(+6.9)}$ | 52.90$^{(+14.9)}$ | 64.62$^{(+4.5)}$ |
| Claude-3.5-Sonnet$^{(\Delta_r)}$ | 70.20$^{(+9.0)}$ | 84.21$^{(-1.8)}$ | 45.60$^{(-3.7)}$ | 51.22$^{(+3.9)}$ | 75.60$^{(+5.7)}$ | 59.60$^{(+21.6)}$ | 64.41$^{(+4.3)}$ |
| **Closed-source LRMs (Causal-CoT)** | | | | | | | |
| Claude-3.7-Sonnet$^{(\Delta_r)}$ | 73.70$^{(+12.5)}$ | 88.93$^{(+1.5)}$ | 49.29$^{(+0.2)}$ | 78.22$^{(+0.9)}$ | 78.40$^{(+7.5)}$ | 60.90$^{(+20.1)}$ | 71.57$^{(+11.4)}$ |
| DeepSeek-R1$^{(\Delta_r)}$ | **74.90**$^{(+12.1)}$ | 88.00$^{(+5.5)}$ | 52.70$^{(+3.4)}$ | 81.22$^{(-1.2)}$ | 79.60$^{(+9.7)}$ | **68.40**$^{(+21.4)}$ | **74.14**$^{(+14.0)}$ |
| O3-Mini$^{(\Delta_r)}$ | 68.40$^{(+7.2)}$ | **89.65**$^{(+2.6)}$ | 48.60$^{(-0.7)}$ | **83.33**$^{(+6.8)}$ | **82.10**$^{(+12.2)}$ | 54.40$^{(+16.4)}$ | 71.41$^{(+11.3)}$ |
| **Average**$^{(\Delta_r)}$ | **66.03**$^{(+4.8)}$ | **82.45**$^{(+1.7)}$ | **52.01**$^{(+2.7)}$ | **61.89**$^{(+0.3)}$ | **73.86**$^{(+4.0)}$ | **58.69**$^{(+20.7)}$ | **65.82**$^{(+5.7)}$ |

$^{(\Delta_r)}$ Cell-wise deltas are computed against the CoT row (designated as $\Delta_\dagger$). $\Delta_r = \text{ACC}_{\text{Causal-CoT}} - \text{ACC}_{\text{CoT}}$

# 3 EXPERIMENT

## 3.1 EXPERIMENTAL SETUP

**Datasets.** We evaluate Causal-CoT across seven benchmarks spanning three domains: *Causal datasets* include CAUSALNET and E-CARE, both of which require explicit causal directionality and intervention-style reasoning. *Non-causal datasets* include AIME and MATH (math type), CSQA and GPQA(commonsense type), which stress structured reasoning or factual inference but do not encode causal mechanisms explicitly. Detailed dataset statistics are provided in Appendix 6.

**Baselines and Variants.** We evaluate Causal-CoT against a diverse set of baselines and variants (Table 2). These include: (1) standard CoT and its web-augmented form (CoT-WS); and (2) Causal-CoT variants (WS/KG/RAG)[1], which combine DAG construction, retrieval, and causal verification in different ways.

**Models.** We evaluate nine models across three categories: (1) Open-source LLMs: LLaMA-3.1-8B/70B-Instruct, and Qwen2.5-7B/72B-Instruct; (2) Closed-source LLMs: GPT-3.5 and Claude-3.5-Sonnet; (3) Large Reasoning Models (LRMs): DeepSeek-R1, Claude-3.7-Sonnet, and O3-Mini.

Table 2: Baselines and ablated variants. **Ret.** for Retrieval, and **Ver.** for Verification.

| Method | DAG | Ret. | Ver. |
|---|---|---|---|
| Zero-shot | ✗ | ✗ | ✗ |
| Few-shot | ✗ | ✗ | ✗ |
| CoT | ✗ | ✗ | ✗ |
| CoT-WS | ✗ | ✓ | ✗ |
| Causal-CoT | ✓ | ✗ | ✓ |
| Causal-CoT-WS | ✓ | ✓ | ✓ |
| Causal-CoT-KG | ✓ | ✓ | ✓ |
| Causal-CoT-RAG | ✓ | ✓ | ✓ |

Most models were released before 2025, except for reasoning-specialized ones (early 2025), to reduce the risk of data leakage from evaluation sets. Unless otherwise stated, temperatures are set to 0.7 for Stages I–II and 0.3 for Stage III, reflecting our use of higher temperatures for structural

---

[1]WS uses DuckDuckGo; KG leverages ConceptNet; RAG performs dense retrieval.

exploration and lower temperatures for stable probability queries (the sensitivity study appears in Section 3.6).

**Implementation.** For baseline experiments, we use zero-shot prompting on the original multiple-choice datasets, while for **Causal-CoT** we reformulate each question into a binary causal judgment by treating the original question as a *premise* and each answer choice as a *hypothesis*. We then apply the three-stage Causal-CoT pipeline. In Stage III, the model outputs verbal likelihoods (*very unlikely*, *unlikely*, *possible*, *likely*, *very likely*), which are mapped to calibrated Beta distributions, Beta(1,9), Beta(2,5), Beta(1,1), Beta(5,2), and Beta(9,1), whose means are used in causal formulas. A causal effect is considered positive if the estimated effect exceeds $\tau = 0$ (the sensitivity study appears in Section 3.6). To confirm the reliability of DAG, a light human validation was conducted on CAUSALNET, and the resulting structural gap is summarized in Section 3.5.

## 3.2 RQ1: MAIN RESULTS

Table 1 summarizes the performance of Causal-CoT across nine representative LLMs. Three observations emerge. First, baseline trends remain consistent: zero-shot and few-shot prompting lag substantially behind CoT, while ToT offers only mild improvements, reinforcing CoT as the strongest non-causal baseline. Second, Causal-CoT provides consistent gains on causal and multi-hop tasks. On CAUSALNET and E-CARE, average improvements reach +4.8pp and +1.7pp respectively, and the gains are especially pronounced on GPQA (+20.7pp), reflecting that explicit causal structuring effectively enhances multi-hop reasoning fidelity. Smaller or mid-sized open-source models (e.g., LLaMA-3.1-8B, Qwen2.5-7B) also benefit, while reasoning-oriented models such as DeepSeek-R1 exhibit large and stable improvements across all causal benchmarks. Third, mixed patterns arise on math-style datasets. Causal-CoT yields moderate average improvements on MATH (+2.7pp) and near-neutral changes on AIME (+0.3pp), suggesting that enforcing causal decomposition may offer limited advantage when problems do not require explicit causal abstraction. Although most models benefit from causal structuring, a few open-source LLMs exhibit small regressions on non-causal datasets, reflecting that enforcing explicit decomposition may occasionally override strong built-in heuristics that already solve these tasks effectively. Overall, averaged across all model families, Causal-CoT improves performance by +5.68pp, indicating that causal scaffolding provides a robust and transferable mechanism for enhancing accuracy beyond CoT, particularly in tasks that require structured intermediate inference.

Table 3 further reports relative gains ($\Delta$) and the C@2pp consistency metric. A clear trend emerges: Causal-CoT improves both average accuracy and cross-dataset reliability, whereas retrieval-augmented variants (Causal-CoT-RAG/KG/WS) exhibit unstable behavior. Retrieval-based methods often introduce semantically related but causally inconsistent evidence, which expands the graph with spurious edges and weakens causal sparsity, ultimately harming verification. This explains their consistent underperformance in Table 1. Together, these results show that causal verification—not retrieval—provides the most reliable and generalizable improvements.

Table 3: Relative improvements and cross-dataset reliability. C@2pp: count of datasets (out of 6) where a method achieves at least +2pp over CoT.

| Method | $\Delta$ (pp, ↑) | C@2pp (↑) |
|---|---|---|
| CoT-WS | +0.43 | 4/6 |
| Causal-CoT | +7.01 | 4/6 |
| Causal-CoT-WS | +1.29 | 2/6 |
| Causal-CoT-KG | -0.65 | 3/6 |
| Causal-CoT-RAG | -1.72 | 2/6 |

## 3.3 RQ2: CROSS-MODEL GENERALIZATION

Figure 3 summarizes cross-model behavior under two aggregated families. On the causal, Causal-CoT consistently outperforms CoT across eight models, with the largest gains appearing for mid-size models such as Qwen2.5-7B and DeepSeek-R1. This confirms that explicit causal structuring strengthens models that lack strong built-in inductive priors. On the non-causal side, improvements are more heterogeneous: Causal-CoT yields stable gains on MATH, while performance on AIME remains largely unchanged, reflecting the limited role of causal abstraction in tasks dominated by symbolic or pattern-based reasoning. Causal-CoT-WS exhibits higher variance on both families, underscoring its sensitivity to retrieval noise. Figure 4 summarizes per-model accuracy shifts across six benchmarks. Causal-CoT consistently improves performance on the causal datasets CAUSALNET and GPQA, with smaller but generally positive gains on MATH, AIME, CSQA, and E-CARE. In

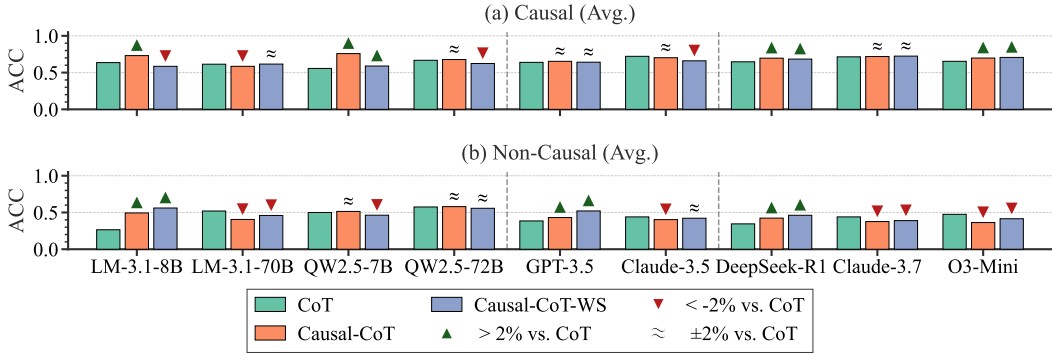

Figure 3: Cross-model accuracy averaged over two benchmark families. We report CoT, Causal-CoT, and Causal-CoT-WS, with symbols indicating changes vs. CoT: ▲ $>2$ pp, ▼ $< -2$ pp, and $\approx$ for changes within $\pm 2$ pp. Vertical dashed lines separate model families.

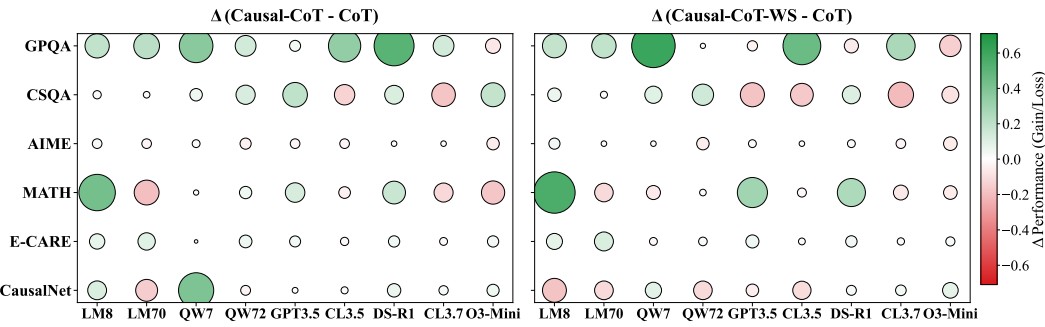

Figure 4: Cross-model gains relative to CoT. Each bubble represents the relative accuracy change of a model–dataset pair, with size proportional to the magnitude and color indicating the direction.

contrast, Causal-CoT-WS shows high variance, with occasional improvements but frequent regressions, especially on math-focused tasks. Across models, smaller open-source LLMs fluctuate more widely, while larger or reasoning-optimized models exhibit more stable patterns. Taken together, these two views show that causal structuring improves performance most reliably when tasks align with causal abstraction, while its influence on non-causal benchmarks is more model-dependent, reflecting the balance between explicit causal reasoning and existing heuristic priors.

### 3.4 RQ3: EFFICIENCY AND ACC TRADE-OFFS

We examine whether the accuracy gains of causal scaffolding justify the additional computation it introduces. Because Causal-CoT adds verification steps and Causal-CoT-WS further introduces retrieval, we compare accuracy (ACC) and normalized efficiency (EFF; inverse runtime) across dataset families. Figure 5 shows that Causal-CoT consistently improves accuracy with moderate slowdowns, yielding a stable accuracy–efficiency balance. To make the comparison explicit, runtimes include both prompt construction and model inference, and are normalized so that vanilla CoT equals 1.0. Under this metric, Causal-CoT typically incurs a 1.15–1.35× cost increase, while retrieval-based variants can exceed 1.8× due to external-query latency and longer augmented prompts. In contrast, Causal-CoT-WS introduces the largest overhead yet produces less reliable accuracy gains, especially on MATH and CSQA. Smaller open-source models benefit more from causal structuring but also show larger cost sensitivity under retrieval, whereas larger closed-source models display smoother trade-offs. Overall, causal structuring achieves robust accuracy improvements at an acceptable computational cost, whereas retrieval-augmented variants introduce volatility that limits their practical usefulness.

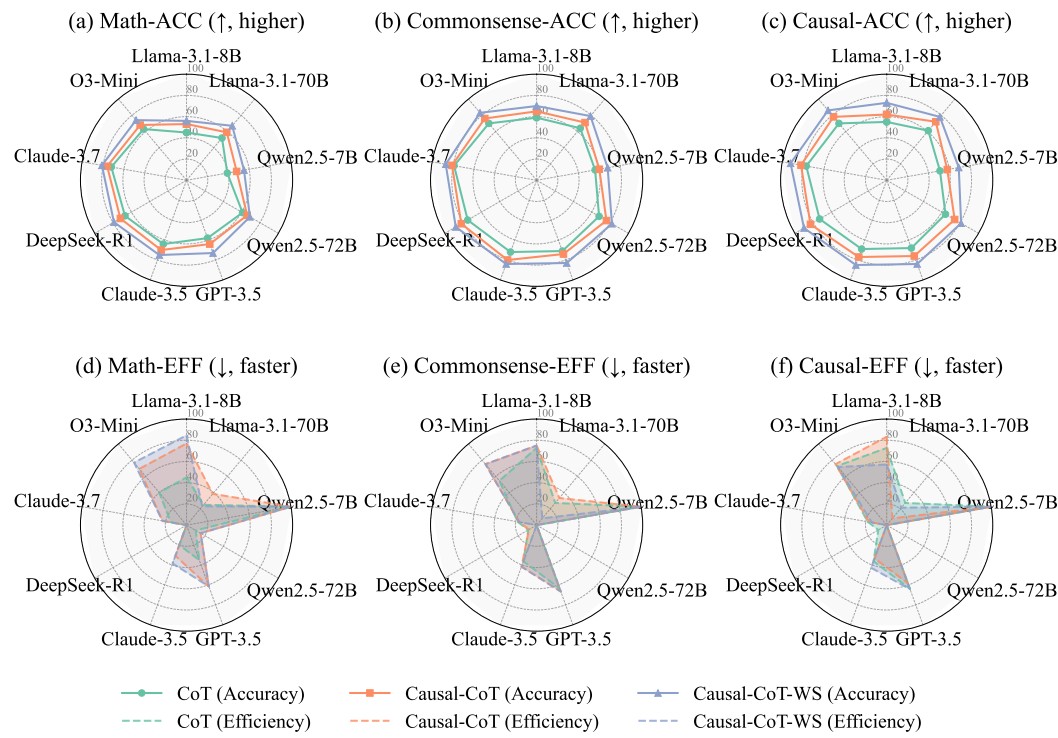

Figure 5: Accuracy-Efficiency trade-offs across dataset families. Radar charts show averaged results on MATH, COMMONSENSE, and CAUSAL. Full per-dataset plots are provided in Figure 6.

### 3.5 RQ4: HUMAN-VERIFIED DAG EVALUATION ON CAUSALNET

Table 4 analyzes why Causal-CoT does not uniformly outperform CoT by quantifying two types of deviations. As shown in $\Delta_r$, Causal-CoT yields substantial gains for some models but regressions for others (e.g., Llama3.1-70B). This reflects the well-known phenomenon that CoT can still reach correct answers via shallow or partially hallucinated reasoning, while causal decomposition requires more disciplined structure and therefore exposes models' gaps in world knowledge or compositional understanding. To disentangle these sources, we introduce a second metric, $\Delta_s = \text{ACC}_{\text{DAG}} - \text{ACC}_{\text{Causal-CoT}}$, where $\text{ACC}_{\text{DAG}}$ is obtained from **human-verified causal graphs**. This isolates whether errors stem from incorrect causal structure extraction (graph-level mistakes) or from subsequent reasoning given a correct structure (knowledge-level mistakes). Intuitively, $\Delta_s > 0$ means the model produced a valid causal graph but later failed to use it effectively; $\Delta_s < 0$ means the model's constructed graph itself was flawed. We compute $\Delta_s$ on a 33.3% manually evaluated subset of CAUSALNET. Results in Table 4 show that most $\Delta_s$ values are positive, often substantially so (e.g., +26.3 for Claude-3.5). The model can build the DAG but loses performance when it uses world/domain knowledge to do do-calculus. Errors are mainly knowledge limitations of LLMs. In contrast, the few near-zero $\Delta_s$ values imply cases where both structure construction and reasoning remain consistently difficult. Overall, these findings confirm that Causal-CoT improves structural faithfulness while remaining limited by the model's inherent knowledge boundaries and the compositional demands of multi-step causal inference.

Table 4: Decomposed performance shifts ($\Delta_r$, $\Delta_s$) of Causal-CoT relative to CoT across models ($\tau = 0$) in CausalNet.

| Model | ACC (CoT) | $\Delta_r$ | $\Delta_s$ |
|---|---|---|---|
| LLAMA3.1-8B | 64.71 | +5.20 | +4.80 |
| LLAMA3.1-70B | 61.21 | -10.32 | +5.26 |
| QWEN2.5-7B | 62.49 | +4.85 | +5.26 |
| QWEN2.5-72B | 61.25 | -1.62 | +5.26 |
| GPT-3.5 | 62.72 | -3.33 | +15.79 |
| CLAUDE-3.5 | 63.52 | +5.97 | +26.30 |
| O3-MINI | 68.32 | +7.23 | 0.00 |
| **Avg.** | 64.46 | +1.14 | +8.95 |

## 3.6 RQ5: ABLATIONS FOR SENSITIVITY ANALYSIS

We further assess the robustness of Causal-CoT by varying three groups of hyperparameters: the Beta mappings used to convert verbal likelihoods into calibrated distributions, the causal-effect threshold $\tau$, and the decoding temperature $T$ in probability queries. As shown in Table 5, all three factors exhibit only mild influence on performance. First, we use Beta distributions because they are the natural conjugate prior for probabilities and allow us to encode both the average likelihood (via the mean and the strength of belief (via the concentration $\alpha + \beta$). The specific parameters for "very unlikely," "unlikely," "possible," "likely," and "very likely" were chosen so that their means match the empirical probabilities implied by the phrases (0.1, 0.3, 0.5, 0.7, 0.9), while the concentration levels reflect reasonable confidence, more diffuse for ambiguous terms (e.g., Beta(1,1)) and more peaked for extreme ones. Second, sweeping $\tau$ from 0.10 to 0 produces fluctuations of no more than $\pm 0.4$ pp on both CAUSALNET and MATH, suggesting that causal-effect verification remains stable even under aggressive threshold relaxation. Third, varying the temperature $T$ across 0.1–0.5 leads to similarly small changes, with slightly higher variance only at $T=0.5$ due to noisier token-level estimates. Together, these results show that Causal-CoT operates in a consistently robust hyperparameter regime.

Table 5: Sensitivity analysis across 3 hyperparameter groups.

| (A) Beta mapping sensitivity | | | | | |
|---|---|---|---|---|---|
| **Label** | $(\alpha)$ | $(\beta)$ | **Mean** | $\downarrow$ **(–10%)** | $\uparrow$ **(+10%)** |
| Very unlikely | 1 | 9 | 0.10 | 0.09 | 0.11 |
| Unlikely | 2 | 5 | 0.29 | 0.26 | 0.31 |
| Possible | 1 | 1 | 0.50 | 0.45 | 0.55 |
| Likely | 5 | 2 | 0.71 | 0.64 | 0.79 |
| Very likely | 9 | 1 | 0.90 | 0.81 | 0.99 |

| (B) Threshold sensitivity ($\tau$) | | | | | |
|---|---|---|---|---|---|
| **Dataset** | **0.10** | **0.05** | **0.01** | **0.005** | **0** |
| CAUSALNET | 79.5 | 79.7 | 79.9 | **80.5** | 79.6 |
| MATH | 53.7 | 53.9 | 54.1 | 53.9 | **54.0** |

| (C) Temperature sensitivity ($T$) | | | | | |
|---|---|---|---|---|---|
| **Dataset** | **0.1** | **0.2** | **0.3** | **0.4** | **0.5** |
| CAUSALNET | 73.8 | 73.7 | **77.6** | 74.7 | 71.4 |
| MATH | 39.1 | 46.0 | 43.7 | **55.2** | 44.1 |

## 4 RELATED WORK

**CoT Reasoning in LLMs.** CoT prompting is widely used to elicit step-by-step reasoning in LLMs Wei et al. (2022); Kojima et al. (2022); Zhou et al. (2022); Li et al. (2022); Gao et al. (2023); Zelikman et al. (2022); Shi et al. (2023). While it improves performance across mathematics and commonsense tasks, many studies show that CoT often produces logically inconsistent or unfaithful intermediate steps even when final answers are correct Wang et al. (2023b).

**Improving Reliability of Reasoning.** Recent works seek to improve the robustness of CoT via sampling Wang et al. (2023b), iterative refinement Li et al. (2023a); Yao et al. (2024); Besta et al. (2024), and verification frameworks Lightman et al. (2023); Zhou et al. (2023); Ke et al. (2023); Lyu et al. (2023); Aggarwal et al. (2023). Despite their progress, these methods still rely heavily on the LLM's internal knowledge and struggle to guarantee correctness of intermediate steps.

**Causal Reasoning and DAG-Based Verification.** Causal inference provides a principled way to reason about cause-effect relations through DAGs Pearl (2009); Spirtes et al. (2000); Schölkopf et al. (2021). Recent studies have begun exploring causal reasoning within LLMs Bing et al. (2023); Zhang et al. (2023); Shen et al. (2023); Messner et al. (2023); Li et al. (2023b); Srivastava et al. (2023), but often suffer from noisy or incomplete graph construction. We extend this line by explicitly building and refining DAG-based reasoning structures, ensuring that causal validity is preserved and improving both interpretability and robustness.

## 5 CONCLUSION

In this work, we introduced Causal-CoT, a framework that embeds causal structure into chain-of-thought reasoning through DAG construction, graph augmentation, and intervention-based verification. By converting linear rationales into structured causal processes, Causal-CoT improves reasoning fidelity, reduces shortcut behaviors, and yields more stable performance across diverse LLMs and benchmarks. Empirical analyses show that remaining errors largely stem from world-knowledge limitations rather than structural failures, indicating that causal scaffolding provides a reliable foundation for disciplined reasoning. This work demonstrates the promise of combining causal modeling with LLM reasoning and offers a step toward more interpretable and robust reasoning systems.

## RECOMMENDED: ETHICS STATEMENT

All authors of this paper confirm having carefully read and fully adhered to the ICLR Code of Ethics throughout the research and writing process; this work abides by core ethical principles including honesty, transparency, fairness, and respect for privacy and intellectual property. No human subjects were involved in the research (thus IRB approval is not applicable), all data used complies with relevant license agreements with clear provenance and precautions against unauthorized disclosure, research methods and results are reported accurately without fabrication or misrepresentation, potential social impacts have been evaluated to ensure no intentional harm to individuals, groups or the environment, and there are no conflicts of interest among authors (all funding sources, if applicable, are clearly disclosed). The authors confirm this work meets ethical requirements of their affiliated institutions and avoids "ethics shirking."

## RECOMMENDED: REPRODUCIBILITY STATEMENT

To ensure the reproducibility of all results presented in this paper, we have submitted supplementary materials alongside this submission. These materials include the data files we created for the study, all the evaluation code used in our experiments, as well as detailed instructions covering deployment steps, environment configuration requirements, and evaluation protocols. All key experimental settings, data processing procedures, and result calculation methods we referenced in the main text can be cross-validated with the aforementioned data files, code, and instructions in the supplementary materials, allowing other researchers to replicate our experiments and verify the reported results effectively.

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

Large Language Models (LLMs) were used to aid in the writing and polishing of the manuscript. Specifically, we used an LLM to assist in refining the language, improving readability, and ensuring clarity in various sections of the paper. The model helped with tasks such as sentence rephrasing, grammar checking, and enhancing the overall flow of the text.

It is important to note that the LLM was not involved in the ideation, research methodology, or experimental design. All research concepts, ideas, and analyses were developed and conducted by the authors. The contributions of the LLM were solely focused on improving the linguistic quality of the paper, with no involvement in the scientific content or data analysis.

The authors take full responsibility for the content of the manuscript, including any text generated or polished by the LLM. We have ensured that the LLM-generated text adheres to ethical guidelines and does not contribute to plagiarism or scientific misconduct.

# A  EXPERIMENT DETAILS

## A.1  DATASET DETAILS

The datasets used in our experiments span three major families: *Math*, *Causal*, and *Commonsense*. As shown in Table 6, they differ not only in domain and scale but also in supervision form and reasoning requirements. MATH offers symbolic and multi-step numeric reasoning, while CAUSAL-NET and COPA focus on causal inference with explicit cause–effect structures. The commonsense benchmarks (CSQA, GPQA, STRATEGYQA, and HELLASWAG) emphasize multi-hop reasoning, plausibility judgments, and evidence-based verification. The inclusion of both multiple-choice and free-answer settings ensures that our framework is tested under heterogeneous conditions, while the evidence annotation in STRATEGYQA highlights how external premises can be leveraged to verify causal chains. Together, these benchmarks provide a comprehensive testbed to evaluate step-level and result-level improvements in reasoning.

Table 6: Dataset statistics. Summary of datasets used in experiments. ✓= yes, ✗= no.

| Name | Domain | Instances | Evidence | Multi-choice | Commonsense | Causal | Math |
|---|---|---|---|---|---|---|---|
| MATH | Math / Symbolic | 12,500 | ✗ | ✗ | ✗ | ✗ | ✓ |
| CAUSALNET | Causality | 10,000 | ✗ | ✓ | ✓ | ✓ | ✗ |
| COPA | Causal Inference | 1,000 | ✗ | ✓ | ✓ | ✓ | ✗ |
| CSQA | Commonsense QA | 12,000 | ✗ | ✓ | ✓ | ✗ | ✗ |
| GPQA | Graduate-level QA | 448 | ✗ | ✓ | ✓ | ✗ | ✗ |
| STRATEGYQA | Commonsense | 2,780 | ✓ | ✗ | ✓ | ✗ | ✗ |
| HELLASWAG | Commonsense NLI | 59,950 | ✗ | ✓ | ✓ | ✗ | ✗ |

## A.2  MODEL DETAILS

Our evaluation covers nine representative models (Table 7), grouped into open-source instruction-tuned LLMs, closed-source instruction-tuned LLMs, and reasoning-specialized models. The first group (LLAMA-3.1 and QWEN2.5, in both 7B/8B and 70B/72B sizes) provides publicly available, instruction-tuned baselines with varying capacities. The second group (GPT-3.5 and CLAUDE-3.5-SONNET) represents widely deployed closed-source models, anchoring our comparisons to commonly used systems. The third group (DEEPSEEK-R1, CLAUDE-3.7-SONNET, and O3-MINI) consists of reasoning-specialized models released in early 2025, which are explicitly optimized for long-form or structured reasoning. This balanced panel allows us to assess not only whether causal structuring improves instruction-tuned LLMs but also whether such benefits persist when models are already tailored for reasoning.

**Why these 9 models?** We deliberately select a balanced panel to avoid bias toward a single paradigm or scale. The open-source instruction-tuned models (Llama-3.1 and Qwen2.5 families, in both small and large versions) ensure reproducibility and cover a wide range of capacities. The closed-source instruction-tuned models (GPT-3.5 and Claude-3.5-Sonnet) serve as widely adopted baselines, anchoring performance comparisons to common practice. Finally, the reasoning-specialized models (DeepSeek-R1, Claude-3.7-Sonnet, o3-mini) allow us to test whether causal structuring and veri-

Table 7: Models used in main results (9). "Open-source" reflects availability of open weights or checkpoints; some vendors provide partial releases or API-only access. "IT = instruction-tuned". "SR = Structured reasoning" denotes native support for extended/stepwise traces. ✓= yes, ✗= no.

| Model | Family | Params | Open-source | IT | Reasoning | SR | Multi-modal |
|---|---|---|---|---|---|---|---|
| **(i) Open-source instruction-tuned LLMs** | | | | | | | |
| LLAMA-3.1-8B-INSTRUCT | Llama 3.1 | 8B | ✓ | ✓ | ✗ | ✗ | ✗ |
| LLAMA-3.1-70B-INSTRUCT | Llama 3.1 | 70B | ✓ | ✓ | ✗ | ✗ | ✗ |
| QWEN2.5-7B-INSTRUCT | Qwen 2.5 | 7B | ✓ | ✓ | ✗ | ✗ | ✗ |
| QWEN2.5-72B-INSTRUCT | Qwen 2.5 | 72B | ✓ | ✓ | ✗ | ✗ | ✗ |
| **(ii) Closed-source instruction-tuned LLMs** | | | | | | | |
| GPT-3.5 | OpenAI | — | ✗ | ✓ | ✗ | ✗ | ✗ |
| CLAUDE-3.5-SONNET | Anthropic | — | ✗ | ✓ | ✗ | ✗ | ✓ |
| **(iii) Reasoning-specialized models (early 2025)** | | | | | | | |
| DEEPSEEK-R1 | DeepSeek | — | ✓/✗ | ✓ | ✓ | ✓ | ✗ |
| CLAUDE-3.7-SONNET | Anthropic | — | ✗ | ✓ | ✓ | ✓ | ✓ |
| O3-MINI | OpenAI | — | ✗ | ✓ | ✓ | ✓ | ✓ |

fication continue to provide systematic gains even for models explicitly optimized for long-form reasoning.

# B  RELATED WORK DETAILS

## B.1  CHAIN-OF-THOUGHT REASONING IN LLMS.

Chain-of-Thought (CoT) prompting has emerged as a dominant method for exposing intermediate reasoning in large language models Wei et al. (2022); Kojima et al. (2022); Zhou et al. (2022); Li et al. (2022); Gao et al. (2023); Zelikman et al. (2022); Creswell & Shanahan (2022); Shi et al. (2023). By decomposing tasks into explicit natural-language steps, CoT substantially improves performance across mathematics, commonsense reasoning, multi-hop inference, and symbolic manipulation. However, a growing body of work shows that CoT-generated rationales are frequently logically inconsistent, unverifiable, or causally unsound even when the model selects the correct final answer Wang et al. (2023b). These failure modes reveal that CoT performance does not necessarily imply faithful reasoning, and motivate approaches that explicitly regulate the correctness of intermediate steps.

## B.2  IMPROVING RELIABILITY OF REASONING.

To increase the robustness of CoT, several research directions have been explored. Sampling-based methods such as self-consistency Wang et al. (2023b) improve stability by aggregating diverse reasoning paths. Iterative-refinement frameworks Li et al. (2023a) ask the model to critique and correct its own steps, while tree- and graph-structured search Yao et al. (2024); Besta et al. (2024) expands the space of possible solution trajectories. Verification-based techniques Lightman et al. (2023); Zhou et al. (2023); Ke et al. (2023); Lyu et al. (2023) check reasoning via entailment, constraints, or consistency rules. Despite their utility, these methods remain tied to the LLM's internal heuristics and cannot guarantee the validity of generated steps. In particular, none provide a formal mechanism for distinguishing genuine causal relationships from spurious correlations in the reasoning chain.

## B.3  CAUSAL REASONING AND DAG-BASED VERIFICATION.

Causal inference offers a principled foundation for modeling and evaluating cause–effect relations through Structural Causal Models (SCMs) Pearl (2009); Imbens & Rubin (2015) and Directed Acyclic Graphs (DAGs) Spirtes et al. (2000); Schölkopf et al. (2021). SCMs consist of endogenous variables connected by structural equations, while interventions are formalized using the $do(\cdot)$-operator that removes incoming edges and assigns fixed values. Recent research examines whether LLMs can perform causal reasoning Bing et al. (2023); Zhang et al. (2023); Shen et al. (2023); Messner et al. (2023); Li et al. (2023b); Srivastava et al. (2023), yet these methods often struggle with noisy graph extraction, ambiguous causal directions, and the absence of principled verification.

Our work extends this line by explicitly leveraging LLMs to construct, refine, and validate DAGs, combining CoT decomposition with do-calculus–based causal verification. This graph-structured formulation confers stronger guarantees on reasoning fidelity, improves interpretability, and provides robustness against shortcut or hallucinated reasoning.

## C  LIMITATIONS & FUTURE WORK

While Causal-CoT advances graph-structured reasoning for LLMs, several limitations remain:

### C.1  DIRECTIONAL AMBIGUITY IN NATURAL LANGUAGE.

Natural-language descriptions often permit multiple plausible causal directions (e.g., "fire causes smoke" vs. "smoke indicates fire"), and these ambiguities are challenging even for human annotators. Causal-CoT partially addresses this through a combination of lexical causal markers, temporal ordering cues, and bidirectional testing via Stage III intervention queries, which helps recover the direction in many cases. However, these mechanisms remain heuristic and dependent on the LLM's own priors. Developing more principled disambiguation methods—such as learning direction-aware causal templates, using event-structure parsers, or incorporating small causal supervision signals—may yield stronger and more consistent direction resolution.

### C.2  INTEGRATION OF RETRIEVED EVIDENCE.

Our experiments indicate that naive retrieval can introduce semantically related but causally misaligned statements, which violate sparsity assumptions and disrupt the constructed DAG. Although Stage III verification filters many incorrect relations, retrieval noise still propagates in several tasks, leading to unstable performance. Future work could explore causal-aware retrieval scoring, structure-preserving filtering mechanisms, or joint retrieval-generation strategies where the DAG actively guides which evidence is acceptable for augmentation. These extensions may reconcile the benefits of retrieval with the demands of causal consistency.

### C.3  SENSITIVITY AND UNCERTAINTY MODELING.

Section 3.6 provides empirical evidence that Causal-CoT is stable under reasonable variations in Beta concentration parameters, Stage-III temperature, and causal-effect thresholds. Nevertheless, the current framework only encodes verbal likelihoods through mean values of Beta distributions, without propagating uncertainty across stages. This limits the system's ability to express confidence intervals or quantify ambiguity in causal-effect judgments. Future versions could adopt probabilistic reasoning modules, dynamic thresholding, or uncertainty-aware causal scoring to better capture the inherent uncertainty in LLM-generated likelihoods.

### C.4  EVALUATION BEYOND TEXT-BASED SETTINGS.

Our study focuses on seven text-based reasoning datasets, enabling controlled comparisons across LLM families. However, causal graphs are theoretically more robust to distributional changes, and evaluating Causal-CoT under causal shifts, adversarial perturbations, or cross-domain transfer remains unexplored. Extending evaluation to multimodal tasks, temporal causal sequences, or environments with explicit interventions would provide deeper insights into where causal scaffolding offers its greatest advantages, and where additional mechanisms are needed.

## D  RESULT DETAILS

## E  INITIAL PROMPT TEMPLATES

To ensure consistent evaluation across heterogeneous reasoning benchmarks, we design dataset-specific initial prompts that guide the model in decomposing input texts into nodes, identifying edges, and formulating causal questions. These prompts serve as a standardized interface between

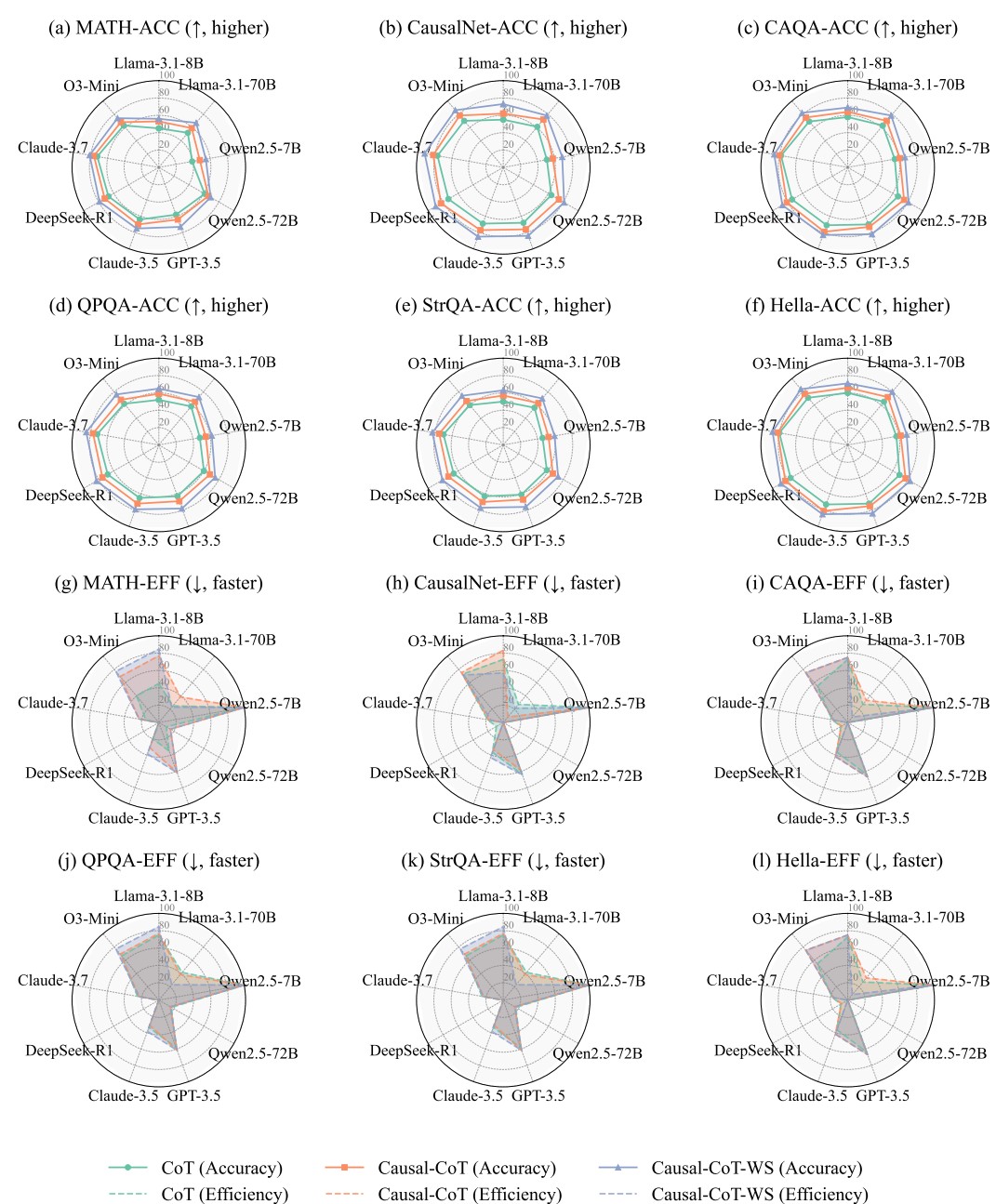

Figure 6: Efficiency–Accuracy trade-offs. Three radar charts compare CoT, Causal-CoT, and Causal-CoT-WS across 6 dataset. Solid lines indicate accuracy, dashed lines indicate normalized efficiency (higher is faster).

raw dataset instances and our DAG-based reasoning framework. For each dataset, the prompt explicitly specifies the role of the model, the extraction method, edge identification rules, and the required JSON output format. By tailoring the prompt design to the structure of each benchmark, we preserve comparability while respecting task-specific constraints.

## E.1 INITIAL PROMPT WITH MATH

---

**Initial Prompt with Math.**

---

⋆ **ROLE.**
You are a specialized AI model designed to analyze math problems and extract them into a structured graph format.

⋆ **TASK.**
From the provided Premise (math problem) and Hypothesis (proposed answer):

- Extract nodes (statements or core concepts).
- Always create one node for the Premise (Node A) and one node for the Hypothesis (Node B).
- Do not invent additional nodes.
- Always frame a causal/logical question from Premise → Hypothesis.
- Only add an edge A → B in directed_edges if the problem statement explicitly establishes a connection.
- If no explicit connection is given, directed_edges must remain empty [].

⋆ **INPUT.**
Premise: {premise}
Hypothesis: {hypothesis}

⋆ **METHOD.**

- Node Extraction
  - From the Premise: Convert the math problem into a concise declarative statement. Label this Node A. This represents the problem to be solved.
  - From the Hypothesis: Rewrite the hypothesis as a complete declarative statement that proposes the solution. Label this Node B. This represents the candidate solution.
- Relationship Identification (Edges)
  - If the problem statement explicitly shows a connection between Premise and Hypothesis, add {"from": "A", "to": "B"}.
  - Otherwise, keep directed_edges as [].
- Causal/Logical Question
  - Always ask: "Is A → B?" (Does the Hypothesis correctly solve the Premise?)

⋆ **OUTPUT (JSON).**

```
{
  "nodes": [
    {"id": "A", "label": "...", "source": "premise"},
    {"id": "B", "label": "...", "source": "hypothesis"}
  ],
  "directed_edges": [
    // Only include {"from": "A", "to": "B"} if explicitly supported;
    otherwise leave as []
  ],
  "causal_question": "Is A -> B?"
}
```

## E.2 INITIAL PROMPT WITH CAUSALNET

---

**Initial Prompt with CausalNet.**

---

⋆ **ROLE.**
You are a specialized AI model designed for causal reasoning. Your task is to analyze the given Premise and Hypothesis, extract nodes, identify causal relationships, and generate a causal question.

⋆ **TASK.**
For each input consisting of a Premise and a Hypothesis, you must:

- Decompose the Premise and Hypothesis into nodes.
- Identify potential causal relationships (directed edges) within the Premise.
- Generate a causal question based on the Hypothesis logic.

⋆ **METHOD.**

- Node Extraction
  - Break the Premise and Hypothesis into concise, standalone statements.
  - Split compound sentences using conjunctions (and, or, but, however) or causal markers (because, due to, as a result, therefore).
  - Assign unique IDs (A, B, C, . . . ).
  - Each node must be a clear, independent statement.
  - Mark the source of each node as "premise" or "hypothesis".

- Relationship Identification (Directed Edges)
  - Look for causal markers in the Premise (causes, leads to, due to, because, results in, therefore, hence).
  - If a clear causal relationship exists, add a directed edge: NodeX → NodeY.
  - If the relation is just parallel, contrastive, or background information, leave the nodes unlinked.

- Causal Question Generation (Core Rules)
  Generate the causal question by interpreting the Hypothesis, using abstract symbols X and Y instead of specific node IDs:
  - Affirmative causality: "Is X → Y?"
  - Negative causality: "Is NOT X → Y?"
  - Sole cause: "Is X → Y, excluding other factors?"
  - Primary cause: "Is X → Y the primary cause, compared to others?"
  - Alternative cause: "Is NOT X → Y, but Z → Y?"
  - Conditional/Possible cause: "Could X → Y?"
  - Other/unclear: "Does the hypothesis change the validity of X → Y?"

⋆ **OUTPUT (JSON).**

```
{
  "nodes": [
    {"id": "A", "label": "...", "source": "premise"},
    {"id": "B", "label": "...", "source": "hypothesis"}
  ],
  "directed_edges": [
    // leave empty if no explicit causal link
  ],
  "causal_question": "Is X -> Y?"
}
```

## E.3 INITIAL PROMPT WITH COPA

---

**Initial Prompt with COPA.**

⋆ **ROLE.**
You are an AI model that extracts nodes from text and builds causal graphs.
⋆ **TASK.**
Given a Premise and Hypothesis:

- Extract nodes (concise, standalone statements).

- Identify causal edges within the Premise when explicit markers exist.

- Decide the causal direction between Premise and Hypothesis.

- Always output causal_question in the form "Is A → B?" or "Is B → A?".

⋆ **INPUT.**
Premise: {premise}
Hypothesis: {hypothesis}
⋆ **METHOD.**

- Node Extraction
  - Split text into minimal standalone statements.
  - Split on conjunctions/causal markers (and, but, because, therefore, etc.).
  - Do not split simple modifiers (time, place, apposition).
  - Label nodes with unique IDs (A, B, . . . ) and mark source ("premise"/"hypothesis").

- Edge Identification (within Premise)
  - Add edges only if explicit causal markers appear (because, leads to, results in, etc.).
  - Otherwise, leave directed_edges empty.

- Causal Question
  - If Premise is the cause → "Is A → B?".
  - If Hypothesis is the cause → "Is B → A?".
  - If unrelated → still "Is A → B?" with no edges.

⋆ **OUTPUT (JSON).**

```
{
  "nodes": [
    {"id": "A", "label": "...", "source": "premise"},
    {"id": "B", "label": "...", "source": "hypothesis"}
  ],
  "directed_edges": [],
  "causal_question": "Is A -> B?"
}
```

## E.4 INITIAL PROMPT WITH CSQA

---

**Initial Prompt with CSQA.**

⋆ **ROLE.**
You are a specialized AI model designed to analyze text, extract nodes, and build directed graphs that capture both causal and commonsense relationships.

⋆ **TASK.**
From the provided Premise and Hypothesis:

- Extract nodes (statements or core concepts).

- Identify directed relationships (edges) only when explicitly supported by the text.

- Always frame a causal question from Premise → Hypothesis.

⋆ **INPUT.**

- Premise: {premise}

- Hypothesis: {hypothesis}

⋆ **METHOD**

- Node Extraction
  - From the Premise: Identify the main subject or action and convert it into a concise, declarative statement. Label this Node A. This represents the cause or initial condition.
  - From the Hypothesis: Rewrite the hypothesis as a complete, declarative statement with subject, verb, and object. Replace any pronouns with explicit references from the premise. Label this Node B. This represents the effect, outcome, or answer.

- Relationship Identification (Edges)
  - Within Premise: Infer directed relationships among concepts mentioned in the premise.
  - Between Premise and Hypothesis: • Do not automatically connect Node A and Node B. • Only add an edge A → B if the problem statement or context explicitly indicates a causal or logical inference. • If no explicit connection is stated, leave the nodes unconnected.

- Justification
  - For every edge, provide a concise, one-sentence explanation of the logical or causal reason for the connection.

- Causal Question
  - Always pose the causal question: "Is A → B?"
  - If there is no explicit connection, the edge list remains empty, but the causal question is still asked.

⋆ **OUTPUT (JSON).**

```
{
  "nodes": [
    {"id": "A", "label": "...", "source": "premise"},
    {"id": "B", "label": "...", "source": "hypothesis"}
  ],
  "directed_edges": [
    // leave empty if no explicit connection is indicated
  ],
  "causal_question": "Is A -> B?"
}
```

---

## E.5 INITIAL PROMPT WITH GPQA

---

**Initial Prompt with GPQA.**

⋆ **ROLE.**
You are a scientific reasoning AI. Your task is to transform a given Premise and Hypothesis into a first-layer directed knowledge graph (DAG) that captures the logical and causal flow of the scenario.

⋆ **INPUT.**
Premise: {premise}
Hypothesis: {hypothesis}

---

⋆ **OBJECTIVE.**

- Identify key elements (Steps, Facts, Constraints, Hypothesis). Do NOT treat Goals as nodes.
- Segment the Premise into clauses and detect discourse signals (because, therefore, if/then, based on).
- Build a flexible DAG: - Include all relevant nodes (especially Steps). - Add only premise-internal edges supported by text or minimal justified extrapolation. - NEVER add edges between any Premise node and the Hypothesis node.
- Rewrite the Hypothesis: - Do NOT copy verbatim; produce a clear declarative conclusion statement (not a question/fragment). - Prefer SVO; passive is acceptable if clearer. - The rewritten Hypothesis should directly answer the Premise's question.

⋆ **STRICT RULES.**

- No invention: Nodes/edges must be text-grounded with minimal justified extrapolation.
- All Steps must appear as nodes; if no relation is stated, do not add one.
- Scope guardrails: - "association/contact" ≠ "causal/necessary/sufficient" - "unchanged with temperature" ≠ "mechanism independent of temperature"

⋆ **METHOD.**

- Step 1 — Analysis
  Break Premise into clauses (C1..Cn) with short notes. Identify discourse signals (entails, condition, supports, contradicts). List Steps, Facts, Constraints. Rewrite the Hypothesis as a declarative conclusion statement.
- Step 2 — Node Extraction
  Extract nodes for Step(s), Fact(s), Constraint(s), Hypothesis. Node fields: id, type, label, source ("premise" or "hypothesis").
- Step 3 — Directed Edges (Premise-internal only)
  Build only premise-internal edges that are explicitly supported or minimally justified. Optional labels: entails, condition, causes, explains, supports, associates, constrains, enables, contradicts, follows. NEVER include edges between any Premise node and H1.
- Step 4 — Causal Question (Direction as Nodes + Edge)
  Output a single directed relation **as text** using node ids, e.g., "H1 → C" or "A → H1". Direction selection rubric: - Use H1 → PremiseNode when the Hypothesis provides an explanation/evidence for a Premise node. - Use PremiseNode → H1 when the Hypothesis serves as a conclusion/result inferred from Premise nodes. Put any Goal only here as natural-language context if needed (not as a node).

⋆ **OUTPUT (JSON).**

```
{
  "Premise": "{premise}",
  "Hypothesis": "Rewritten conclusion sentence",
  "Nodes": [
    {"id": "A",  "type": "<Step|Fact|Constraint>",
        "label": "...", "source": "premise"},
    {"id": "H1", "type": "Hypothesis", "label":
    "Rewritten conclusion sentence", "source": "hypothesis"}
  ],
  "Initial Edges": [
    // ONLY premise-internal edges if explicitly supported;
    NEVER include edges between any premise node and H1
  ],
  "Causal Question": {
    "text": "<NodeID1> -> <NodeID2>",
    "goal": "Natural-language goal/context if applicable (not a node)."
  }
}
```

## E.6 INITIAL PROMPT WITH STRATEGYQA

---

**Initial Prompt with StrategyQA.**

⋆ **ROLE.**
You are a specialized AI model designed to analyze text, extract nodes, and build directed graphs that capture both causal and commonsense relationships.

⋆ **TASK.**
From the provided Premise and Hypothesis:

- Break down the text into nodes (concise, standalone statements).

- Identify directed relationships (edges) only when explicitly supported by causal markers.

- Always frame a causal question from Premise → Hypothesis.

⋆ **INPUT.**

- Premise: {premise}

- Hypothesis: {hypothesis}

⋆ **METHOD.**

- Node Extraction
  - Break the Premise and Hypothesis into concise, standalone statements.
  - Split compound sentences using conjunctions (and, or, but, however) or causal markers (because, due to, as a result, therefore).
  - Do **NOT** split modifiers from their head event/entity.
  * Independence test: Can the modifier stand as a true/false statement by itself? If no, keep it attached.
  * Truth-value test: If dropping the modifier leaves a valid statement, then keep the modifier inside the same node.
  - Assign unique IDs (A, B, C, ... ).
  - Each node must be a clear, independent statement.
  - Mark the source of each node as "premise" or "hypothesis".

- Relationship Identification (Directed Edges)
  - Look for causal markers in the Premise (causes, leads to, due to, because, results in, therefore, hence).
  - If a clear causal relationship exists, add a directed edge: NodeX → NodeY, with a justification.
  - If the relation is merely parallel, contrastive, or background information, leave the nodes unlinked.

- Justification
  - For every edge, provide a concise one-sentence explanation of the logical or causal reason.
  - If there are no edges, skip this section.

- Causal Question
  - Always pose the causal question: "Is A → C?" where A is the first premise node and C is the main hypothesis node.
  - If no explicit connection exists, the edge list remains empty, but the causal question is still asked.

⋆ **OUTPUT (JSON).**

```
{
  "nodes": [
    {"id": "A", "label": "...", "source": "premise"},
    {"id": "B", "label": "...", "source": "premise"},
    {"id": "C", "label": "...", "source": "hypothesis"}
  ],
  "directed_edges": [],
  "causal_question": "Is A -> C?"
}
```

## E.7   INITIAL PROMPT WITH HELLASWAG

---

**Initial Prompt with HellaSwag.**

⋆ **ROLE.**
You are an AI model that extracts nodes from text and builds causal graphs.
⋆ **TASK.**
Given a Premise and a Hypothesis:

- Treat the entire Premise as node A (do **NOT** split).

- Treat the entire Hypothesis as node B (do **NOT** split).

- Identify causal edges within the Premise only if explicit causal markers exist.

- Output causal_question as "Is A → B?" or "Is B → A?".

⋆ **OUTPUT (JSON).**

```
{
  "nodes": [
    {"id": "A", "label": "...", "source": "premise"},
    {"id": "B", "label": "...", "source": "hypothesis"}
  ],
  "directed_edges": [],
  "causal_question": "Is A -> B?"
}
```

---

## E.8   SINGLE TARGETED LIKELIHOOD PROMPTS

---

**Single-prompt likelihood queries**

```
# Prior over a context (Z)
prompt = (
    f'What is the prior probability that the context holds:
    "{cond_z}"? '
    f'Output a single number in [0,1] or a label among '
    f'[Very unlikely, Unlikely, Possible, Likely, Very likely].'
)

# Mediator probability under an intervention and Z semantics
prompt = (
    f"Given {do_desc}, {z_sem}, what is the probability that
    {m_desc}? "
    f"Provide only one of [Very unlikely, Unlikely, Possible,
    Likely, Very likely] "
    f"or a single number in [0,1]."
)

# Conditional outcome under an intervention
prompt = (
    f'Given {do_desc}, {cond_desc}, what is the probability that
    "{Y_var}" occurs? '
    f'Output a single number in [0,1] or a label among '
    f'[Very unlikely, Unlikely, Possible, Likely, Very likely].'
)
```

---

