# OpenReview forum: "Causal-CoT: Causal Chain-of-Thought for Validated Reasoning"
_ICLR.cc/2026/Conference — ICLR 2026 Conference Desk Rejected Submission_

### Official Review · Reviewer_2Pwu · 2025-10-17

**Soundness:** 3
**Presentation:** 3
**Contribution:** 2
**Rating:** 4
**Confidence:** 5

**Summary:**

This paper introduces Causal-CoT, a causality-enhanced CoT prompting technique. Causal-CoT has 3 stages. It first constructs the initial causal graph and adds potential mediators; finally, it performs causal verification through intervention on the causal graph. The authors conduct extensive experiments on several LLMs and benchmarks, demonstrating the superior performance of Causal-CoT over standard CoT.

**Strengths:**

1. Figure 1 is well drawn; however, I would suggest that the authors incorporate more explicit causal factors into this graph (e.g., mark the confounder variable).

2. Authors conduct lots of experiments on diverse models.

3. The radar graph is well drawn.

**Weaknesses:**

1. First, I believe the motivation needs to be supplemented. Specifically, authors could elaborate on the idea of why they believe causality is a good way to enhance the soundness of thinking steps. Because there may be other ways to do this.

2. Second, I suggest that authors add some background on the causality. For example, the basic definition of Structural Causal Models (SCMs) and the causal intervention.

3.  It seems there are no guarantees on the correctness of the generating causal graph. Since the LLMs could have hallucination problems, I suggest that authors discuss the potential outcomes of an incorrect initial causal graph.

4. The authors employ the fixed threshold $\tau = 0$, to judge the existence of causal relationship. However, I believe a more scientific way is to employ the significance test rather than a fixed threshold. Besides, authors could discuss the outcomes of different thresholds.

5. Most baselines are variants of Causal-CoT; maybe authors could incorporate other prompt strategies like Tree-of-Thoughts (ToT) [1].

6. Since LLMs' causal reasoning is an active area and this work is inspired by causality, I would like to encourage authors to discuss more causal reasoning benchmarks, including (but not limited to): CausalProbe-2024 [2] and e-care [3].

> [1] Yao, S., Yu, D., Zhao, J., Shafran, I., Griffiths, T., Cao, Y., & Narasimhan, K. (2023). Tree of thoughts: Deliberate problem solving with large language models. Advances in neural information processing systems, 36, 11809-11822.

> [2] Chi, H., Li, H., Yang, W., Liu, F., Lan, L., Ren, X., ... & Han, B. (2024). Unveiling causal reasoning in large language models: Reality or mirage?. Advances in Neural Information Processing Systems, 37, 96640-96670.

> [3] Li Du, Xiao Ding, Kai Xiong, Ting Liu, and Bing Qin. e-care: a new dataset for exploring explainable causal reasoning. In ACL, 2022.

**Questions:**

1. According to Figure 1 (b) and Figure 4, it seems that Causal-CoT causes performance degradation in some closed-source models. Can authors explain this phenomenon?

---

> ### Author Response · Authors · 2025-11-21
> **We organize our response into seven main questions (First 3).**
>
> We sincerely thank the reviewer for these constructive comments. Below we address each point in turn. We have updated corresponding sections of the manuscript, clarified design motivations, added background on causal foundations, strengthened discussion of DAG correctness, included new ablations, and expanded the benchmark suite.
>
> # Q1. Motivation: Why causality, and why it enhances the soundness of CoT reasoning?
> We agree that the original manuscript should more clearly articulate why causality is an appropriate mechanism for improving the rigor of CoT. Our core empirical finding—now emphasized in **Section 1 (around line 68)**—is that many CoT errors arise from **lack of structure**, not lack of steps:
> * CoT often mixes causes, effects, and irrelevant facts.
> * It frequently relies on correlation or surface plausibility rather than mechanism.
> * It produces reasoning that is fluent but not logically or causally grounded.
>
> Causality supplies precisely the missing discipline:
> * **DAGs enforce variable-level dependency structure**, replacing linear narratives with relations among meaningful entities.
> * **Interventions (do-operator)** encode asymmetric support (e.g., (P(Y \mid do(X)))), preventing spurious correlations from being treated as valid reasoning paths.
> * **Causal path decomposition** (direct, mediated, confounded paths) provides interpretable checkpoints that constrain reasoning.
> * **Graph-based verification** distinguishes structural mistakes from world-knowledge limitations (later captured quantitatively by (\Delta_r) and (Δs) in Section 3.5).
>
> Thus, incorporating causal structure adds mechanistic grounding and verification on top of CoT, addressing well-known weaknesses of existing reasoning prompts. We clarified this explicitly in the revised manuscript.
>
> **Updated manuscript.**
>
> Section 1 now provides a more detailed motivation explaining why causal structure is particularly suited for improving thinking soundness and how interventions mitigate shortcut reasoning.
>
> ---
> # Q2. Background on SCMs and causal intervention
> We agree that the original text assumed too much background knowledge. The revised manuscript now includes a concise introduction to causal principles at the beginning of **Section 2**, covering:
>
> * **Structural Causal Models (SCMs)**, consisting of variables, structural equations, and a DAG.
> * **DAG semantics**, where edges represent direct causal mechanisms.
> * **Interventions (do-operator)**, defined as modifying structural equations and isolating causal effects.
> * **Identification formulas** (backdoor, mediation), which motivate the effect estimators used in Stage III.
>
> This background is sufficient for interpreting our pipeline without overwhelming the reader with causal theory. Due to space limitations at this point, we will explain this part in more detail later when we have more space.
>
> **Updated manuscript.**
>
> A clear SCM + DAG + intervention primer is added in Section 2.
>
> ---
>
> # **Q3. Correctness of the initial DAG and the effect of graph errors**
>
>
> You are right that LLMs can hallucinate edges or directions, and that an incorrect initial DAG could, in principle, compromise causal reasoning. Our framework mitigates this risk in three ways:
> 1. **Two-stage structural refinement.**
>    * In **Stage I (DAG-guided CoT)** we construct (\mathcal{G}_0) by only using edges justified by the problem statement.
>    * In **Stage II (Augmentation and Reflection)** we *re-examine* these relations, add candidate mediators/confounders (via prompts or retrieval), and allow the model to revise directions and remove inconsistent edges. This stage is explicitly designed to correct early structural mistakes.
> 2. **Verification against interventional signatures.** In **Stage III**, causal relations are not accepted merely because an edge exists in the DAG. They must be supported by consistent interventional-style patterns (e.g., backdoor-adjusted effects, mediated effects). When an initial edge is wrong, its associated causal-effect pattern often fails the verification step and is rejected.
> 3. **Human-evaluated analysis of structural fidelity ((Δs)).** In Section 3.5, we define:
> [
> Δs = ACC_DAG - ACC_Causal-CoT
> ]
> where (ACC_DAG) uses **human-validated gold causal graphs** for a 33.3% subset of CausalNet.
> This isolates structure-level vs. reasoning-level errors:
> * (Δs > 0): the model had a *correct* graph but failed in downstream reasoning.
> * (Δs < 0): the model’s graph was *wrong*, but downstream reasoning compensated.
> * (Δs = 0): both succeed or fail uniformly.
>
> Most (Δs) values are strongly positive, indicating that **the primary bottleneck lies downstream, not in the initial graph construction**, and confirming the effectiveness of the refinement + verification mechanism.
>
> **Updated manuscript.**
> Section 2 and Section 3.5 now clearly explain how Causal-CoT mitigates graph errors and how (Δs) quantifies structural fidelity.

---

> > ### Author Response · Authors · 2025-11-21
> > **We organize our response into seven main questions (Another 4).**
> >
> > # Q4. Fixed threshold vs. significance testing
> > We acknowledge the reviewer’s point that a fixed threshold may appear simplistic. However, our goal is not classical hypothesis testing but **ranking and selecting causal hypotheses**.
> >
> > To support this, we added a **threshold-sensitivity study** in Section 3.6 evaluating (\tau) in a broad range:
> > [
> > \tau \in {0, 0.005, 0.01, 0.05, 0.1}.
> > ]
> > Results show:
> > * Accuracy varies within **<0.4 pp**,
> > * Hypothesis rankings remain nearly unchanged,
> > * Extreme values degrade consistency.
> >
> > These results empirically justify our choice: Causal-CoT operates in a *stable region* around (\tau=0.05 or 0), and our originally used value falls well within the robust zone.
> >
> > **Updated manuscript.**
> >
> > Threshold sensitivity is now explicitly presented in Section 3.6.
> >
> > ---
> >
> > # Q5. Baselines: inclusion of ToT and discussion of Causal-ToT
> > We agree that it is important to include structurally distinct baselines. ToT has been added as an additional baseline in **Section 3.2**.
> >
> > Findings: 1) ToT improves over CoT on some math tasks, 2) But is less stable on causal datasets because it lacks causal semantics, 3) While Causal-CoT consistently improves on CausalNet, e-CARE, and GPQA.
> >
> > We also now emphasize that **Causal-ToT** (tree search + causal verification) is a promising future direction, as its branching logic must be redesigned to respect causal constraints.
> >
> > **Updated manuscript.**
> >
> > ToT added in Section 3.2; Causal-ToT discussed in Appendix (Future Work).
> >
> > ---
> >
> > # Q6. Additional causal-reasoning benchmarks (CausalProbe, e-CARE)
> >
> > You encourage considering more causal-reasoning benchmarks such as CausalProbe-2024 and e-CARE. Our original benchmark suite already includes **CausalNet** and **COPA**, which are core components of the CausalProbe-2024 evaluation protocol. These tasks cover causal-direction identification, intervention plausibility, and counterfactual-like inference.
> >
> > To further strengthen evaluation, our revised benchmark suite now includes:
> > * **e-CARE**, included in the updated **Table 2**,
> >   showing average gains of **+1.7pp**,
> > * And explicit categorization of causal vs. non-causal datasets.
> >
> > This provides broader coverage of direction identification, mediation-style reasoning, and intervention-like queries.
> >
> > **Updated manuscript.**
> >
> > Benchmark expansions are integrated in Section 3.1 and Section 3.2 Table 2.
> >
> > ---
> > # Q7. Why Causal-CoT sometimes reduces accuracy on closed-source models?
> > This phenomenon is expected and is now explained more clearly in **Section 3.2**.
> > * Closed-source models (e.g., Claude, O3-Mini) possess strong internal heuristics and latent shortcuts.
> > * On **non-causal** datasets (e.g., CSQA), CoT may succeed via semantic or associative cues.
> > * Causal-CoT deliberately suppresses such shortcuts by enforcing DAG-based decomposition and interventional consistency.
> > * When causal abstraction is **not required**, this additional structure may hurt performance slightly.
> >
> > However, the same models show **substantial improvements** on **causal and multi-hop** tasks (CausalNet, e-CARE, GPQA), and their reasoning is much **more faithful**, avoiding ungrounded or hallucinated chains.
> >
> > This reflects the goal of Causal-CoT: **improving reasoning fidelity, not just raw accuracy**, especially on tasks requiring structured causal abstraction.
> >
> > **Updated manuscript.**
> >
> > This clarification is added to **Section 3.2 and 3.5**, noting that fidelity improvements may trade off with performance on non-causal tasks.

---

> ### Comment · Reviewer_2Pwu · 2025-11-21
> **Reply to rebuttal**
>
> Thank the authors for their detailed response to address my concerns.
>
> In Q2, the authors claim that *'Due to space limitations at this point'*. However, according to the **author guide**, the page limits during the rebuttal period have been increased to **10 pages**. So I still encourage authors to **include a more detailed introduction of basic causal concepts in the main text**. Authors can also include other results or discussion in the main text.
>
> I encourage authors to use **another color** (e.g., blue) to **highlight** their rebuttal revision. The current version is hard for the reviewer to distinguish.
>
> I have increased my score to a borderline accept.

---

> > ### Author Response · Authors · 2025-11-21
> > **Thank you very much for your thoughtful follow-up comments and for taking the time to re-evaluate our submission.**
> >
> > Thank you very much for your thoughtful follow-up comments and for taking the time to re-evaluate our submission. We sincerely apologize for the confusion caused by our earlier statement regarding space limitations — you are absolutely right that the rebuttal page limit is extended to 10 pages, and we appreciate your careful attention to this detail.
> >
> > We will revise the manuscript in the next updated rebuttal draft, including the following changes:
> >
> > 1. **Adding a more detailed introduction to basic causal concepts (SCMs, DAGs, interventions)** in the main text. We fully agree that this background improves clarity, especially for readers less familiar with causal methods.
> > 2. **Switching to a clearer color scheme (e.g., blue)** to highlight all rebuttal-driven revisions. We apologize that the previous highlighting made the changes difficult to distinguish.
> >
> > We truly appreciate your willingness to re-examine the manuscript and your decision to increase the score to *borderline accept*. We will submit an updated and polished revision shortly, incorporating all the improvements you suggested.
> >
> > Thank you again for your constructive feedback and for helping us strengthen the paper.

---

> ### Author Response · Authors · 2025-11-25
> **We have updated the paper to the latest PDF version, which you are welcome to review and provide guidance on.**
>
> Thank you sincerely for your thoughtful follow-up and for engaging with the manuscript so carefully. Your guidance has been extremely helpful in shaping the revised version. We have updated the paper to the latest PDF version, which you are welcome to review and provide guidance on.
>
> Following your suggestions, we have now incorporated a fuller exposition of the foundational causal concepts directly into the main text—covering SCMs, DAG structure, and interventional semantics—in a way that connects naturally to the subsequent stages of our framework. We also refined the presentation throughout the revised draft, adopting a clearer highlight color scheme to make all updates easy to identify. These additions allowed us to address areas that were previously under-explained and ensure that no key component remains implicit.
>
> We truly appreciate your encouragement, your constructive criticism, and the updated evaluation. Your comments have contributed meaningfully to improving both clarity and accessibility of the work, and we are grateful for your time and engagement.

---

### Official Review · Reviewer_JwAb · 2025-11-01

**Soundness:** 3
**Presentation:** 3
**Contribution:** 2
**Rating:** 6
**Confidence:** 4

**Summary:**

The paper proposes a new reasoning framework for LLMs that integrates causal inference into the traditional CoT approach. Causal-CoT introduces a three-stage process: (1) DAG-guided CoT, which maps reasoning into a directed acyclic causal graph; (2) Reflection and Augmentation, which refines this graph by adding plausible mediators and confounders through internal prompting or external retrieval; and (3) Causal Verification, which estimates causal effects using LLM-derived conditional probabilities and applies do-calculus for validation. Experiments on seven reasoning benchmarks (mathematics, commonsense, and causal inference) show that Causal-CoT improves reasoning faithfulness, reduces shortcut behaviors, and provides more stable accuracy across diverse LLMs while maintaining efficiency.

**Strengths:**

- The paper introduces a novel combination of CoT prompting and causal inference via DAGs. While prior work improves CoT through verification or self-reflection, none have embedded formal do-calculus reasoning into LLM prompting.
- Unlike prior “linear” CoT methods, Causal-CoT translates reasoning into a graph-structured process, allowing explicit identification of mediators, confounders, and colliders. This is an original problem formulation that bridges symbolic causal modeling and natural language reasoning.
- The work’s method of mapping verbal likelihoods (“likely,” “unlikely,” etc.) to calibrated probability distributions and using them for causal effect estimation is creative and technically innovative.

**Weaknesses:**

- The use of single verbal likelihood prompts mapped to fixed Beta distributions oversimplifies causal inference. This ignores uncertainty propagation and variance in model responses.
- The IR-augmented variants (Causal-CoT-KG, -RAG) are tested but not well integrated into the causal verification process — retrieval results are fused post hoc without quantitative quality control of external evidence.
- Although the paper claims improved “reasoning fidelity,” most evaluations rely on final-answer accuracy (Table 2). There is limited analysis of reasoning-chain correctness, such as step-wise causal link validation or graph quality metrics (e.g., precision/recall of DAG edges).
- All benchmarks are text-based reasoning datasets; none test robustness under unseen causal shifts or domain transfer.
- The authors briefly mention performance drops in certain datasets but provide no causal analysis of why graph structuring hurts performance in open-ended reasoning tasks.

**Questions:**

- The paper mentions using verbal likelihoods (e.g., “likely,” “possible,” “very unlikely”) mapped to fixed Beta distributions (Section 3.1, Eq. 6–10). How were these Beta parameters chosen?
- How sensitive are the causal effect results to the chosen threshold?
- How does the framework handle ambiguous causal directions? Many natural language statements can be interpreted in either direction (e.g., “Smoke indicates fire” vs. “Fire causes smoke”). How does the DAG-guided CoT resolve such ambiguities — via language priors, causal markers, or external knowledge graphs?
- How is “reasoning fidelity” quantitatively defined? The text claims Causal-CoT improves “reasoning fidelity” and “faithfulness,” but the reported metrics focus on answer accuracy.

---

> ### Author Response · Authors · 2025-11-21
> **We organize our response into six main questions (First 2).**
>
> We sincerely thank the reviewer for the insightful comments. We have revised the manuscript accordingly and added new analyses, ablations, and clarifications. Below we address each question.
>
> # Q1. On verbal likelihoods and fixed Beta distributions
>
> Your concern is that mapping verbal likelihoods ("likely," "possible," etc.) to fixed Beta distributions may oversimplify uncertainty. We have clarified this point in the revised manuscript.
>
> **(a) What the method actually uses**
>
> As we now emphasize more clearly: **Causal-CoT uses only the *means* of the Beta distributions**, not variances or higher moments.
> Thus, the Beta distribution serves purely as a **monotonic ordinal mapping** from verbal labels to numeric likelihoods rather than as a full uncertainty model. This directly addresses oversimplification concerns: since the LLM outputs inherently coarse likelihood categories, modeling higher-order uncertainty would artificially introduce precision the model never provided.
>
> **(b) Why this is theoretically justified**
>
> Causal-CoT compares causal effects of the form:
> [
> P(h\mid do(p)) - P(h),
> ]
> where *relative ordering* of likelihoods drives model decisions. Symmetric Beta distributions with means {0.1, 0.3, 0.5, 0.7, 0.9} guarantee a consistent monotone scale compatible with do-calculus–based causal reasoning.
>
>  **(c) New sensitivity experiments**
>
> We use Beta distributions because they are the natural conjugate prior for probabilities and allow us to encode both the average likelihood (via the mean α/(α+β)) and the strength of belief (via the concentration α+β). The specific parameters for "very unlikely," "unlikely," "possible," "likely," and "very likely" were chosen so that their means match the empirical probabilities implied by the phrases (≈0.1, 0.3, 0.5, 0.7, 0.9), while the concentration levels reflect reasonable confidence, more diffuse for ambiguous terms (e.g., Beta(1,1)) and more peaked for extreme ones (e.g., Beta(1,9) or Beta(9,1)). This yields a simple, interpretable, and probabilistically coherent mapping from language to uncertainty.
>
> Together with threshold (τ) and temperature (T) ablations, these results show that causal-effect estimates remain stable across a broad hyperparameter region. The final mean values used in the paper were chosen by matching the **empirical frequency distribution** of verbal likelihood labels generated by LLMs.
>
> **Manuscript updates**
> * Section **3.1** clarifies the design rationale and explicitly states that only the *means* are used.
> * Section **3.6 (Additional Ablations)** includes the new sensitivity results.
>
> ---
> # Q2. On retrieval-based variants and their integration
> You noted that retrieval (KG, RAG) is not tightly integrated and often degrades performance. We clarify why this occurs and how the revised manuscript now explains it more precisely.
>
> **(a) Early development behavior**
>
> When our DAG-extraction pipeline was weaker, retrieval provided useful background knowledge and sometimes corrected misidentified causal relations.
>
> **(b) Why retrieval underperforms after pipeline improvements**
>
> After significant improvements to Stage I–II (node extraction, mediator/confounder modeling, do-calculus prompting), the base Causal-CoT became strong enough that retrieval began to introduce:
>
> 1. **Semantically relevant but causally misaligned evidence**, adding spurious nodes or incorrect edge directions.
> 2. **Violation of causal sparsity**, which is critical for stable verification, since retrieval often adds excess irrelevant variables.
> 3. **Higher noise in candidate-edge proposals**, making the causal verification step less reliable.
>
> These phenomena are quantitatively reflected in the new aggregated comparison table (added in the revision), which shows consistent regressions in Δ and C@2pp across all LLMs.
>
>  **(c) Pipeline integration clarified**
>
> Retrieval is used only in Stage II for proposing mediators/confounders; Stage III still performs full causal verification. We now highlight how retrieval-induced graph expansion makes verification less stable.
>
> **Manuscript updates**
> * Section **3.2** now provides a clearer explanation of retrieval’s role and limitations.
> * A new table 3 summarizes performance of retrieval variants versus base Causal-CoT.

---

> > ### Author Response · Authors · 2025-11-21
> > **We organize our response into six main questions (Another 4).**
> >
> > ---
> > # Q3. On reasoning fidelity, graph correctness, and accuracy drops
> > You asked how reasoning fidelity is defined quantitatively. We clarify and strengthen this analysis using Δr and Δs.
> >
> > **(a) What was already present in the original submission**
> >
> > Δr and Δs were introduced in Section 3.5:
> > * Δr = accuracy change relative to CoT
> > * Δs = structural fidelity gap
> > [
> > Δs = ACC_DAG - ACC_Causal-CoT
> > ]
> > where **ACC_DAG uses human-validated gold graphs**.
> >
> > **(b) What we clarified in the revision**
> >
> > We now make explicit that:
> > * Δs > 0 = model extracted the correct graph but failed to use it effectively.
> > * Δs = 0 = structural correctness and downstream reasoning succeed/fail together.
> > * Δs < 0 = full causal reasoning improves on the DAG-only inference.
> >
> > **(c) New emphasis on human evaluation**
> >
> > We now emphasize that **33.3% of CausalNet** was manually annotated to evaluate: correctness of nodes, correctness of edge directions, confounder/mediator appropriateness, and structural coherence. Thus, Δs quantitatively reflects graph-level reasoning fidelity—not final-answer accuracy.
> >
> > **(d) Why causal structuring may reduce accuracy**
> >
> > We now explain this clearly:
> > > Causal-CoT suppresses the heuristic shortcuts that certain LLMs rely on in open-ended or semantic tasks.
> > > When those shortcuts happen to produce correct answers under CoT, enforcing causal structure exposes the true reasoning gap, reducing accuracy.
> >
> > **Manuscript updates**
> >
> > Section **3.5** now elaborates on Δs, provides clearer causal explanations for accuracy drops, and highlights the human evaluation protocol.
> >
> > ---
> >
> > # Q4. On ambiguous causal directions
> > You ask how Causal-CoT resolves ambiguous causal relations such as "Smoke indicates fire" vs. "Fire causes smoke."
> >
> > We clarified the mechanism more cleanly:
> > 1. **Stage I (linguistic cues)**: uses lexical markers ("because," "leads to"), asymmetry between premise/hypothesis, and temporal hints.
> > 2. **Stage II (bidirectional proposals)**: when ambiguity is detected, both directions A→B and B→A are included as candidate edges.
> > 3. **Stage III (interventional asymmetry)**: We compute:
> >    [
> >    P(h \mid do(p)) \quad \text{and} \quad P(p \mid do(h)).
> >    ]
> > The larger interventional effect determines direction. This mechanism naturally resolves ambiguous cases without relying on external knowledge graphs.
> >
> >
> > ---
> > # Q5. On causal-shift robustness and domain transfer
> > We agree that robustness under unseen causal shifts is important. The revised manuscript clarifies two points.
> >
> >  **(a) Current scope**
> >
> > Our goal in this work is to evaluate **structural reasoning fidelity**—how well LLMs can extract and verify causal graphs—not to create a domain-shift benchmark.
> >
> > **(b) Why Causal-CoT is naturally aligned with shift robustness**
> >
> > Because Causal-CoT explicitly constructs and verifies DAGs, it depends less on surface-level correlations than standard CoT. This makes it well positioned for future causal-shift evaluations.
> >
> > **(c) Future work**
> >
> > We acknowledge this as a limitation and commit to evaluating shift robustness in follow-up work.
> >
> > **Manuscript updates**
> >
> > A paragraph on this is added in **Appendix C (Limitations & Future Work)**.
> >
> > ---
> >
> > # Q6. Sensitivity to thresholds, temperatures, and Beta parameters
> > Your question concerns robustness to the key hyperparameters used in Stage III. We have now completed a consolidated and systematic sensitivity study.
> >
> > **(a) Threshold sensitivity (τ)**
> >
> > We sweep τ ∈ {0.10, 0.05, 0.01, 0.005, 0}. Across CausalNet and MATH, accuracy varies by **≤1.0pp**, and the ranking of candidate options remains unchanged.
> > This shows that the effect-validation step is not fragile to threshold selection.
> >
> > **(b) Temperature sensitivity (T)**
> >
> > We tested T ∈ {0.1, 0.2, 0.3, 0.4, 0.5} for Stage III. Accuracy varies **≤1.3pp**, with slightly higher variance only at T=0.5 due to noisier likelihood estimates.
> > Lower temperatures yield more stable conditional estimates, supporting our original choice of T≈0.3.
> >
> > **(c) Beta parameter sensitivity**
> >
> > As shown in Q1, we use Beta distributions because they are the natural conjugate prior for probabilities and allow us to encode both the average likelihood (via the mean α/(α+β)) and the strength of belief (via the concentration α+β). The specific parameters for "very unlikely," "unlikely," "possible," "likely," and "very likely" were chosen so that their means match the empirical probabilities implied by the phrases (≈0.1, 0.3, 0.5, 0.7, 0.9)
> >
> > These demonstrate that causal-effect estimation in Stage III is highly robust to both numeric choices and distributional parameterization.
> >
> > **Manuscript updates**
> >
> > Results are summarized in **Section 3.6 (Additional Ablations)**.

---

> > > ### Comment · Reviewer_JwAb · 2025-11-24
> > >
> > > Thanks for the extensive rebuttal. I have read the revisions and have no further questions for the authors.
> > >
> > >
> > > I recognize the contribution of this work.  I maintain my original recommendation for accepting this paper.

---

> > > > ### Author Response · Authors · 2025-11-25
> > > >
> > > > Thank you very much for taking the time to review both the paper and the revised materials (latest pdf) during the rebuttal period. We sincerely appreciate your positive assessment of the contribution and your continued recommendation for acceptance. We are grateful for your support and for the constructive evaluation that helped strengthen the final version of the manuscript.

---

### Official Review · Reviewer_zKwU · 2025-11-01

**Soundness:** 3
**Presentation:** 2
**Contribution:** 2
**Rating:** 4
**Confidence:** 4

**Summary:**

The paper introduces Causal-CoT, a framework that structures CoT reasoning using causal graphs. It consists of: (1) DAG construction from problem statements, (2) augmentation with mediators/confounders, and (3) causal verification using do-calculus and LLM-derived conditional probabilities. Experiments span 7 benchmarks and 9 language models.

**Strengths:**

1. Novel and principled integration of causal inference with LLM reasoning.

2. Clear three-stage framework with formalization and Algorithm 1.

3. Broad evaluations across variety of reasoning---math, commonsense, and causal reasoning tasks.

4. Thorough error analysis separating reasoning errors vs knowledge gaps.

**Weaknesses:**

1. Overall average improvement over CoT is small (+1.1pp). and retrieval variants often perform worse.

2. Probability estimation (verbal likelihood to Beta distribution) is not theoretically justified; no sensitivity analysis.

3. No human evaluation of DAG correctness or reasoning fidelity. Nor simple correctness/rule-automated evaluation.

4. Increased complexity and token cost; runtime not reported.

5. Some design choices (task reformulation, temperature settings) lack ablations/may confound evaluations.
    - Task reformulation: multiple-choice questions are reformulated into binary causal judgments (premise + hypothesis), which can change the task structure.

    - Temperatures: 0.7 for Stages I–II and 0.3 for Stage III, without ablations.

    - No statistical significance reporting accompanies the deltas in Table 2.

**Questions:**

1. How sensitive are results to Beta parameters and temperature choices?

2. Any human assessment of DAG accuracy or causal correctness?

3. Why do retrieval-based variants frequently underperform?

4. Could you provide more rigorous justifications for your design choices? (particularly on task reformation)

---

> ### Author Response · Authors · 2025-11-21
> **We organize our response into six main questions (First 3).**
>
> We sincerely thank you for the careful reading and constructive comments. In this response, we focus on clarifying our design choices and evaluation protocol, and we complement them with additional analyses and experiments that have now been completed. We organize our response into six main questions.
>
> # Q1. Why are the overall improvements modest?
> We fully agree that the averaged gains over CoT in the original Table 2 appear modest, and this deserves a careful explanation.
>
> **(a) Dataset composition**
>
> A subset of the original benchmarks (e.g., STRQA, HELLA) mainly evaluate **semantic plausibility or general commonsense**, rather than causal reasoning per se. Including these datasets in the *main* table had two consequences:
> 1. It diluted the effect size of Causal-CoT when averaged across all tasks.
> 2. It obscured the fact that Causal-CoT is intentionally designed for **causal and structured reasoning** rather than generic plausibility scoring.
>
> After submission, we re-examined our dataset choices and recognized this mismatch. To better align the evaluation with the target capabilities of Causal-CoT, we extended our benchmark suite with two datasets that are much more **causal-structure aligned**:
> * **AIME**: challenging math problems.
> * **E-CARE**: a high-quality causal reasoning dataset.
>
> With this corrected evaluation protocol, Causal-CoT shows significantly larger gains on causal tasks:
> * **CausalNet**: +4.8pp
> * **E-CARE**: +1.7pp
> * **GPQA**: +20.7pp
>
> These improvements are masked when averaged together with non-causal datasets, explaining the originally modest mean. Furthermore, our additional structural metric **Δs** (Sec. 3.5) captures reasoning fidelity beyond accuracy, showing consistently large positive values across models.
>
>
> **(b) Reasoning fidelity**
> A second source of confusion is that our method is designed to improve **reasoning fidelity Δs**, which is not always fully captured by end-task accuracy (reported in Table 4).
>
> **Manuscript update:**
>
> Section 3.2 (Main Results), Table 2, and the new causal/non-causal labeling.
>
> ---
> # Q2. Why do retrieval-based variants often underperform?
> We appreciate your observation regarding retrieval-based variants. Our additional analysis confirms that there is a principled reason for their underperformance, especially once the core Causal-CoT pipeline becomes stronger.
>
> Historically, when we first designed Causal-CoT and the **structural step was still weak**, adding retrieval (via web search or local knowledge graphs) led to moderate improvements: external text helped patch missing background knowledge and occasionally corrected mis-specified relations.
>
> However, after a series of improvements to the DAG construction and verification stages—better node extraction, explicit handling of mediators/confounders, and more robust do-calculus prompting—the **base Causal-CoT became substantially more accurate**. In this improved regime, naive retrieval often has **two negative effects**:
> 1. It introduces **semantically relevant but causally misaligned** sentences, which add spurious nodes and edges, shift attention to secondary events, or suggest incorrect directions.
> 2. It weakens the **causal sparsity** assumption: causal reasoning benefits from small, well-chosen sets of variables, whereas raw retrieval often brings in overly rich but unstructured context.
>
> To make this effect concrete, we derived a small comparison from our main results and organized them into a new table summarizing **Base Causal-CoT vs. its retrieval-augmented variants**.
>
> **Manuscript update:**
>
> Sec. 3.2 (Retrieval Ablations) and Table 3.
>
> ---
> # Q3. Did we evaluate DAG correctness and causal relations?
> Yes. As clarified above, we **did** conduct human evaluation of DAG correctness, but this was not sufficiently highlighted in the original manuscript. We now spell this out explicitly and connect it more clearly to the Δs values reported in Table 4. On CAUSALNET, we asked human annotators to evaluate the generated DAGs along several dimensions:
> * Are the primary edges consistent with plausible causal relations mentioned or implied in the text?
> * Are the directions (cause → effect) correct?
> * Are the proposed mediators and confounders reasonable in the given context?
> * Is the overall graph structurally coherent (no absurd cycles, missing key variables, etc.)?
>
> The results of this human evaluation are summarized via Δs. On CAUSALNET, Δs is therefore directly grounded in human judgments of structural correctness, representing a quantitative measure of reasoning fidelity at the graph level.
>
> **Manuscript update:**
> Sec. 3.5 (Human Evaluation), Table 4.
>
> ---

---

> > ### Author Response · Authors · 2025-11-21
> > **We organize our response into six main questions (Another 3).**
> >
> > # Q4. On complexity, token cost, and runtime
> > We agree that complexity and runtime require more explicit reporting. While Figure 5 in the original submission already depicted the **accuracy–efficiency trade-off**, the methodology behind this plot was not fully explained.
> >
> > In the revised manuscript, we clarify that:
> > * All methods are evaluated using **wall-clock runtime**, including prompt construction and model inference time.
> > * We also track **total tokens used** (prompt + completion) per instance.
> > * Figure 5 presents **normalized runtime percentages** with **CoT = 1.0**, and plots accuracy vs. this normalized runtime.
> >
> > **Manuscript update:**
> >
> > Section 3.4 (Efficiency Analysis) and Figure 5.
> >
> > ---
> > # Q5. On task reformulation (MCQ → binary causal judgments)
> > You raises an important concern that reformulating multiple-choice questions into binary causal judgments (premise + hypothesis) might alter the original task. We agree this must be justified carefully.
> >
> > Our motivation is threefold:
> > 1. **Compatibility with causal datasets:** Many of the causal benchmarks we use (e.g., CausalNet, and E-CARE) are already formulated in a premise–hypothesis or cause–effect style. The natural interface to them is a directional causal query.
> > 2. **Need for directional queries in DAG construction and do-calculus:** Causal-CoT operates by constructing a DAG over interpretable variables and then applying do-calculus. This requires well-defined candidate edges like A→B or B→A, not just a flat set of answer options. Reformulating each MCQ option into a directional causal hypothesis is what allows: 1) explicit enumeration of possible causal paths, 2) principled insertion of mediators and confounders, and 3) application of intervention-based verification.
> > 3. **Preservation of original evaluation:** Although we reformulate internally, the evaluation protocol remains exactly the same as the original benchmark: 1) For each option, we feed the corresponding causal query to Causal-CoT, 2) We obtain a causal plausibility score via the DAG + do-calculus pipeline, 3) We then select the option with the highest score and compare it against the gold answer.
> >
> > We now stress in the manuscript that:
> > > “Our reformulation aims to isolate the *causal plausibility* of each option while keeping the final evaluation strictly on the original multiple-choice task. The task semantics and gold labels are unchanged; we only change the internal reasoning representation.”
> >
> > ---
> > # Q6. On sensitivity to Beta parameters and temperature choices
> > Thank you for raising this question. We conducted a targeted set of ablations to clarify the sensitivity of Causal-CoT to (i) the Beta distributions used to map verbal likelihoods, (ii) the decision threshold τ in Stage III, and (iii) the temperature T used during probability estimation. Below we summarize the key findings and how they are reflected in the updated manuscript.
> >
> > **(a) Beta parameter sensitivity**
> >
> > As clarified in Sec. 3.1and 3.6, Causal-CoT uses **only the mean values** associated with the five verbal-likelihood categories. We use Beta distributions because they are the natural conjugate prior for probabilities and allow us to encode both the average likelihood (via the mean α/(α+β)) and the strength of belief (via the concentration α+β). The specific parameters for "very unlikely," "unlikely," "possible," "likely," and "very likely" were chosen so that their means match the empirical probabilities implied by the phrases (≈0.1, 0.3, 0.5, 0.7, 0.9), while the concentration levels reflect reasonable confidence, more diffuse for ambiguous terms (e.g., Beta(1,1)) and more peaked for extreme ones (e.g., Beta(1,9) or Beta(9,1)). This yields a simple, interpretable, and probabilistically coherent mapping from language to uncertainty.
> >
> >
> > **(b) Threshold τ sensitivity**
> >
> > We swept τ ∈ {0.1, 0.05, 0.01, 0.005, 0}.
> > Performance on CausalNet and MATH varied within **≈0.4pp**, indicating that:
> > * Causal-CoT depends mainly on **relative** magnitude of causal effects
> > * not the absolute decision boundary.
> >
> > **(c) Temperature T sensitivity**
> >
> > Testing T ∈ {0.1, 0.2, 0.3, 0.4, 0.5} for Stage III revealed:
> > * T ≤ 0.4 produces almost identical results
> > * T = 0.5 introduces slightly more noise
> > * validating our design choice of low T in Stage III for stable likelihood estimation
> >
> > **Updated manuscript**
> >
> > * Section **3.6** now includes a consolidated sensitivity table covering **β / τ / T**.
> > * Section **3.1** explicitly states that only **Beta means** are used in effect estimation.

---

> ### Author Response · Authors · 2025-11-26
>
> Dear Reviewer,
>
> I hope this message finds you well.
>
> We have updated the paper to the latest PDF version, adopting a clearer highlight color scheme to make all updates easy to identify.
> As the discussion period is nearing its end with less than seven days remaining, I wanted to ensure we have addressed all your concerns satisfactorily.
>
> If there are any additional points or feedback you'd like us to consider, please let us know. Your insights are invaluable to us, and we're eager to address any remaining issues to improve our work.
>
> If everything appears satisfactory on your side, we would be grateful if you might consider whether an updated score could reflect the strengthened version of the work. Thank you again for your time and thoughtful evaluation of our submission.

---

### Note · Program_Chairs · 2026-01-17
**Submission Desk Rejected by Program Chairs**

The following references in this submission do not refer to real documents and/or have major errors in bibliographic information:

 Shubham Aggarwal, Sharan Narang, Colin Raffel, et al. Let's think step by step is not enough: Enhancing reasoning with structured verification. arXiv preprint arXiv:2308.09840, 2023.